# A new approach for measuring the carbon and oxygen content of atmospherically-relevant compounds and mixtures

James F. Hurley[1], Nathan M. Kreisberg[2], Braden Stump[3], Chenyang Bi[1], Purushottam Kumar[1], Susanne V. Hering[2], Pat Keady[3], Gabriel Isaacman-VanWertz[1]

[1] Department of Civil and Environmental Engineering, Virginia Tech, Blacksburg, VA, 24061
[2] Aerosol Dynamics Inc., Berkeley, CA, 94710
[3] Aerosol Devices Inc., Fort Collins, CO, 80524

*Correspondence to:* Gabriel Isaacman-VanWertz (ivw@vt.edu)

**Abstract**. Due to its complexity, gas- and particle-phase organic carbon in the atmosphere are often classified by their bulk physicochemical properties. However, there is a dearth of robust, moderate-cost approaches to measure bulk chemical composition of organic carbon in the atmosphere. This is particularly true for degree of oxygenation, which critically affects the properties and impacts of organic carbon, but for which routine measurements approaches are lacking. This gap has limited understanding of a wide range of atmospheric components, including particulate matter, the mass of which is monitored worldwide due to its health and environmental effects, but the chemical characterization of which requires relatively high capital costs and complex operation by highly trained technical personnel. In this work, we demonstrate a new approach to estimate the mass of carbon and oxygen in analytes and mixtures that relies only on robust, moderate-cost detectors designed for use with gas chromatography. Organic compounds entering a flame ionization detector were found to be converted with approximately complete efficiency to $CO_2$, which was analysed downstream using an infrared detector to measure the mass of carbon analysed. The ratio of FID signal generated per $CO_2$ formed (FID/$CO_2$) was shown to be strongly correlated ($R^2 = 0.89$) to the oxygen-to-carbon ratio (O/C) of the analyte. Furthermore, simple mixtures of analytes behaved as the weighted average of their components, indicating that this correlation extends to mixtures. These properties were also observed to correlate well with the sensitivity of the FID estimated by structure activity relationships (quantified as the relative Effective Carbon Number). The relationships between measured FID/$CO_2$, analyte O/C, and FID sensitivity allow estimation of one property from another with <15% error for mixtures and <20% error for most individual analytes. The approach opens the possibility of field-deployable, autonomous measurement of the carbon and oxygen content of particulate matter using time-tested, low-maintenance detectors, though such an application would require some additional testing on complex mixtures. With some instrumental modifications, similar measurements on gas-phase species may be feasible. Moreover, potential expansion to additional gas chromatography detectors may provide concurrent measurement of other elements (e.g. sulphur, nitrogen).

 **1 Introduction**

Reactive organic carbon (ROC) in the atmosphere oxidizes in the presence of other natural and anthropogenic emissions to produce aerosols and ozone, air pollutants with significant health, ecosystem, and climate effects (Intergovernmental Panel on Climate Change, 2013; World Health Organization, 2016). ROC is complex, comprised of thousands of different compounds possessing a broad range of structures, volatility, and reactivity (Goldstein and Galbally, 2007). The vapor pressures of ROCs range from highly volatile gaseous compounds to essentially non-volatile species that are emitted in condensed form as primary organic aerosols (POA) or are produced through oxidation of gases to form secondary organic aerosol (SOA) (Donahue et al., 2006; Jimenez et al., 2009; Zhang et al., 2007). Similarly, solubility ranges from essentially insoluble (e.g., volatile hydrocarbons) to highly soluble polyols (Hodzic et al., 2014; Raventos-Duran et al., 2010). Given these broad ranges, there is a keen interest in classifying organic compounds by understanding their bulk physicochemical properties (Donahue et al., 2006; Kroll et al., 2011). A major parameter by which ROC is frequently classified is some measure of the degree of oxidation or oxygenation of each compound (Donahue et al., 2011; Kroll et al., 2011; Pankow and Barsanti, 2009). This is because the oxygen content of a molecule influences its volatility (Aiken et al., 2008; Huffman et al., 2009), its ability to absorb into existing particle-phase (Donahue et al., 2011), and its solubility (Raventos-Duran et al., 2010). Consequently, a robust approach to measuring O/C and properties of organic carbon would be a valuable tool for understanding the impacts of ROC.

Degree of oxygenation particularly affects the impacts of particle-phase organic carbon. Atmospheric particulate matter ("aerosols") is responsible for a substantial fraction of annual global deaths (Dockery et al., 1993; Lim et al., 2012; World Health Organization, 2016) and is consequently monitored, primarily on a mass basis, throughout the world. A major fraction of these aerosols is comprised of organic compounds, which may be directly emitted or form through the atmospheric oxidation of naturally and anthropogenically emitted volatile organic compounds. Owing to the wide range of sources and formation chemistry of aerosols, the composition may vary substantially, and the impacts of aerosols are dependent on their composition. For example, increased oxygenation increases hygroscopicity (Massoli et al., 2010), which increases cloud formation and impacts albedo (Intergovernmental Panel on Climate Change, 2013). Oxidized compounds are also more likely to fragment and volatilize upon further oxidation (Lambe et al., 2012). Furthermore, while aerosols are known to have deleterious effects, the "dose-response curve" that defines the increased risk of a given adverse health impact per unit exposure is poorly constrained (Apte et al., 2015; Burnett et al., 2014; Marshall et al., 2015). This uncertainty may be driven in part by observations that toxicity is compositionally dependent, with certain components of organic aerosol exhibiting higher toxicity than others (specifically: oxygenated compounds (Tuet et al., 2016; Verma et al., 2015), oxidation products of biogenic gas-phase precursors (Kramer et al., 2016; Lin et al., 2016), and biomass burning emissions (Rohr and McDonald, 2016; Verma et al., 2015)). However, regulations and monitoring networks for aerosol mass typically do not include frequent or automated measurements of aerosol composition, limiting understanding of the physical, chemical, and physiological impacts of aerosols. New approaches are needed to facilitate low-maintenance measurements of the composition of aerosols and atmospherically-

relevant compounds, specifically with a focus on oxygen content (e.g., oxygen-to-carbon ratio, O/C), which is a major driver of its health and climate impacts.

Currently, most widely used measurements of aerosol chemical composition use filter-based measurements, in which samples are collected for offline analyses conducted in a lab. Besides the time-integrated (multiple days or weeks) nature of the sampling, another drawback is the delay in the analysis, during which reaction or decomposition of the sample may occur. In contrast, real-time chemical composition data may be obtained with advanced mass spectrometric and/or chromatographic instrumentation such as the Aerosol Mass Spectrometer (AMS), Aerosol Chemical Speciation Monitor (ACSM) (DeCarlo et al., 2006; Ng et al., 2011), or the Thermal-desorption Aerosol Gas chromatograph (TAG) (Williams et al., 2006). However, these instruments are difficult (though not impossible, (see (Budisulistiorini et al., 2013) ) to deploy for long-term operation because of their high capital and operational costs, need for skilled operators, and complex data analysis. Due to these limitations, routine measurements of aerosol composition are primarily limited to quantification of their mass concentration. A small amount of chemical information is available using moderate-cost instrumentation such as the OC/EC analyser (Sunset Labs), which separately quantifies elemental and organic carbon, but no tools are available to provide continuous measurement of the chemical composition of the organic component of aerosols despite the critical role it plays on aerosol impacts. Consequently, there remains a need for new methods that provide at least some chemical information about particle composition (e.g., bulk O/C or properties of individual components) without the need for mass spectrometry or other high-cost or high-complexity techniques.

Moderate-cost robust measurement of aerosol carbon and aerosol constituents has been previously achieved using the operating principle of the flame ionization detector (FID), a common detector used in gas chromatography and previously implemented for bulk particle measurements in OC/EC analysers. In an FID, analytes are combusted in a hydrogen flame, and signal is produced by electrometrically measuring ions (primarily $CHO^+$) produced by the flame (Holm, 1997, 1999). This approach has high sensitivity, a large linear dynamic range ($10^7$ ), and robustness against variations in flow rates since FID is mass-sensitive rather than concentration-sensitive (Skoog et al., 2017) ; consequently, FIDs are sometimes operated in parallel with higher-complexity detectors to achieve low-uncertainty quantification. Most importantly, FID signal is nearly universal, with response proportional to the mass of carbon entering the detector; however, addition of oxygenated functional groups decreases FID response. The impact of oxygen on FID response is overcome in current bulk aerosol instrumentation relying on this detector by catalytically converting all organic carbon to $CO_2$ and then $CH_4$, ensuring universal response to all aerosol carbon (Chow et al., 2001). For applications in which molecular structures of individual analytes are known (e.g., quantification by chromatographic instruments), the FID sensitivity of an analyte can be calculated from established structure-activity relationships (Scanlon and Willis, 1985). Generally, a carbon with a carbonyl does not produce any FID signal, and a carbon with a hydroxyl group produces half as much FID signal as a hydrocarbon. This relationship can be quantified more precisely as the Effective Carbon Number (ECN) of a compound, which describes FID response as equivalent to a hydrocarbon of a certain carbon number. Operating at ambient pressures and requiring only a source of hydrogen, the FID is consequently a stable and low-cost alternative to mass spectrometry, providing robust quantification but with substantially reduced chemical

resolution. However, accounting for the impact of oxygen-containing functional groups on FID response currently relies on either knowing molecular structures or catalytic conversion of all carbon.

Prior work has shown analytes to be combusted highly efficiently in an FID, even for highly oxygenated compounds (Fock, 1976) . This fact, coupled with the general trend that oxygenated functional groups decrease FID sensitivity, suggests that the simultaneous measurement of FID signal and the $CO_2$ produced in the flame should provide some estimate of oxygen content. In this work, we test this hypothesis by measuring the $CO_2$ produced by an FID using a non-dispersive infrared absorption (NDIR) sensor. $CO_2$ detection by NDIR (specifically at wavelength 4.255 µm, wavenumber 2350 $cm^{-1}$) (LI-COR, 2007; Pandey and Kim, 2007) is widely used for continuous field measurements due to its accuracy and stability (Pandey and Kim, 2007). Simple implementation of this method requires only measurement of absolute absorption in a single optical cell, while more accurate (but complex) instruments may include a reference cell, with $CO_2$ measured from the differential absorption between the cells. This latter configuration finds more use in continuous monitoring instruments since any drift or variation in beam strength can be accounted for with the reference measurement (Skoog et al., 1996). The high accuracy afforded by a two-cell approach (levels of detection of less than 100 ppb, (LI-COR, 2007) provides potential detection of FID-produced $CO_2$ at concentrations relevant to atmospheric applications. One significant potential application of using combined FID and $CO_2$ measurements to measure oxygen content would be the measurement of bulk chemical properties of organic compounds. As a preliminary assessment of the feasibility of such an approach, we provide in the Supplementary Information (Section S1 and Table S1) an estimate of the concentrations of $CO_2$ generated in the combustion of atmospherically-relevant concentrations of organic aerosols. Using reasonable assumptions for a theoretical instrument that collects and analyses organic aerosol based on this approach, expected concentrations of $CO_2$ measured in the outflow of an FID of ~100 ppb to ~100 ppm, well within the detection range of current NDIR-based $CO_2$ instrumentation. We focus here on organic aerosols due to their major atmospheric consequences and the known effects of bulk oxygen content on these consequences, but note that this approach could also be adapted to collected samples of gas-phase organic compounds.

Besides oxygen, nitrogen and sulphur are common heteroatoms occurring in atmospheric samples, particularly organic aerosols. Both elements are present at concentrations approximately an order of magnitude lower than oxygen or carbon (Aiken et al., 2008; Carrasquillo et al., 2014; Docherty et al., 2011; Surratt et al., 2008), with organic sulphate primarily present as organosulphates, and nitrogen present as a mixture of functional groups including nitrates (Farmer et al., 2010), N-containing heterocycles (Laskin et al., 2015), and amines (Murphy et al., 2007). These compounds also likely efficiently combust in an FID, but the impacts of these functional groups on FID sensitivity of an analyte are poorly studied. The presence of these groups may complicate the relationship between FID response and oxygen content, just as it increases uncertainty in existing measurements of the oxygen content of bulk aerosol (Farmer et al., 2010). Fortunately, the low concentrations of heteroatoms relative to oxygen and carbon in atmospheric aerosols and aerosol components suggests that their overall impacts on most measurement approaches are relatively minor in most atmospheric applications. Consequently, the focus of the present work is on understanding and parameterizing the fundamental relationships between FID signal, $CO_2$ produced in an FID flame, and oxygen content of the analyte(s), and the impacts of other heteroatoms are left for discussion as uncertainties. There are a

number of potential atmospherically-relevant applications of these relationships, including calibration of instruments for which molecular structures of analytes are not known, and bulk analysis of oxygen content of ambient or laboratory-generated organic aerosol.

In this work, we couple a $CO_2$ detector downstream of an FID to demonstrate a new approach to measure the carbon and

oxygen content of atmospherically-relevant organic compounds. To provide a viable approach, three criteria need to be met:

1) The FID must reproducibly (and ideally completely) combust all organic compounds, converting it to $CO_2$ for detection downstream to provide carbon mass.

2) FID response per carbon atom (i.e., measured ratio of signals, FID/$CO_2$) must be inversely proportional to the oxygen content of analytes and mixtures.

3) One measurable parameter (e.g., FID/$CO_2$ signal ratios, analyte O/C, and analyte FID sensitivity) must predict any other to within reasonable error.

We systematically test these three criteria in this work, with the major goal of being able to predict any one parameter from the others. While this work focuses on the theory and fundamental validation of the underlying approach, we discuss issues to consider when applying this approach to atmospheric samples and propose potential applications to the study of atmospheric

chemistry.

## 2 Materials and Methods

### 2.1 Theory of operation

Previous work has focused on quantifying the FID response of individual compounds, typically quantified as their Effective Carbon Number (ECN). The ratio of this number to the number of carbons, $N_C$, in a compound is termed here the "relative

Effective Carbon Number" (rECN = ECN/$N_C$) and describes the average response of each carbon in a compound relative to hydrocarbon response. The rECN can be thought of as the per-carbon FID sensitivity relative to the maximum possible. Because oxygenated functional groups decrease FID response, the per-carbon FID sensitivity (i.e., rECN) necessarily decreases with increasing oxygen content. The rate of this decrease is dependent on the structure of the oxygenated functional groups because carbonyl and hydroxyl groups do not have equal effects on ECN. In Figure 1, the theoretical slopes of compounds

comprised solely of different substituent functional groups are shown (dashed lines) as a function of O/C, estimated from existing structure-activity relationships (Scanlon and Willis, 1985). Compounds entirely comprised of carbonyls and carboxyls provide bounding cases, as the carbon atom in both groups produces no FID signal and they add one and two oxygen atoms respectively; the slopes of these relationships are consequently rECN = -O/C and rECN = -0.5*O/C, respectively. Compounds comprised only of alcohols fall in between, with a slope dependent somewhat on the specific structures of the alcohols due to

differences in the effects of primary, secondary, and tertiary alcohols (compounds used to estimate slopes shown in Figure 1

are provided in the Supplementary Information: Section S3, Table S4, and Figure S1). ECNs for compounds spanning a range of functionalities are available in published literature and are included in Figure 1. The structure-activity relationships used to calculate rECN, and example calculations are given in Supplementary Information Section S2. Nearly all compounds fall within the theoretical bounding cases as expected, with an average slope approximately in the middle. These data suggest that a direct measurement of rECN of a compound should generally correlate with its O/C.

## 2.2 Instrument description

A schematic of the instrument used to quantify the relationship between FID signal and $CO_2$ produced is shown in Figure 2a. Individual analytes or mixtures were introduced to a thermally controlled cell from which they were thermally desorbed into a helium carrier flow. Analytes were then transferred through heated transfer lines to the FID, using compressed $CO_2$-free zero air as an oxygen source. For some configurations, an in-line GC provided separation of analytes as discussed below. A custom adapter was built to allow tubing to connect the FID outlet to the $CO_2$ detector. To prevent the water produced by the FID hydrogen flame from impairing the carbon dioxide detector, the outflow of the FID was dried using an in-line Nafion permeation dryer (MD-110-48F, Perma Pure LLC) with dry sheath air provided by a zero air generator used as dessicating counterflow. The dried sample stream was detected by a NDIR $CO_2$ analyser (LI-6262 or LI-7000, LI-COR Biosciences), with $CO_2$-free air provided to the reference cell. For all trials, helium served as the carrier and FID make-up gas where applicable. In summary, the required supporting gases as operated here include $CO_2$-free zero air (as FID oxygen source and LI-COR reference), zero air for desiccating counterflow, hydrogen (as FID fuel source), and helium (as carrier gas and FID make-up gas, when necessary). Most, or all, of the required air could be supplied as compressed room air with some minor technical adaptations, reducing consumable gas needs to hydrogen and carrier gas. Theoretically, a flow split prior to the FID enables this configuration to be extended to additional detectors (e.g., flame photometric detector, nitrogen phosphorous detector) as shown in Figure 2a.

Three instrument configurations were used in order to analyse a wide range of compounds:

System 1 (Agilent Injection Inlet - GC separation - Agilent FID - LI-COR 6262): Aliquots of 0.2-1.0 μL of mixtures (including authentic standards, and commercially available fragranced consumer products) were injected into a heated inlet on a gas chromatograph (7890B, Agilent Technologies) and separated by gas chromatography (GC) on a non-polar column (Restek Rxi-5sil MS, 30 m x 0.25 mm x 0.25 um, temperature ramp of 6 °C/s from 40 to 310 °C). Analytes were detected by an on-board Agilent FID. For identification of unknown analytes in mixtures, parallel injections were performed on a GC coupled to a quadrupole mass spectrometer (GC-MS) as a detector (7820A/5977, Agilent Technologies) using the same GC stationary phase, carrier gas, and flow conditions; differences in retention times due to differences in temperature ramps were corrected for by daily injections on each system, of a mixture of $n$-alkanes ($C_7$-$C_{40}$, Supelco). Positive identification of analytes in a

mixture of unknowns was defined as a match in the NIST mass spectral library with strength of at least 850, as well as a retention time in within the range of retention indices published by NIST (Wallace, 2019).

System 2 (Agilent Injector Inlet (2a) or Passivated TAG Cell (2b) - Guard Column - Agilent FID - LI-COR 7000) : Solutions of individual analytes were injected into either a heated inlet (System 2a) or a room temperature passivated metal cell (System 2b). Aliquots of 0.2-1.0 μL individual organic compounds were injected at concentrations of ~500 ng/uL in water or a carbonaceous solvent. Analytes were thermally desorbed and transferred through an inert guard column (Restek Hydroguard, 5m x 0.25 mm) isothermally heated (300 ºC) within the GC oven to the FID. Carbonaceous solvents were separated from the

analyte by either the use of a cryotrap on System 2a or, for System 2b, desorbed at low temperature prior to higher-temperature desorption of the analyte. The cryotrap was situated on the transfer line within the oven and cooled by liquid nitrogen to a temperature that trapped the analyte but not the solvent, then heated to volatilize the analyte of interest.

System 3 (Passivated TAG cell - SRI FID - LI-COR 7000): Individual analytes were injected into a room temperature passivated metal cell as above (System 2b). Solvent was allowed to evolve at slightly elevated temperatures, then the analyte

was thermally desorbed with 20 sccm helium (controlled by a mass flow controller, flow range 100 sccm, Alicat Scientific) through a custom isothermally heated (250° C) transfer line directly to the FID. On this system the FID used was from Scientific Research Instruments (SRI) on a Model 110 Detector Chassis, with FID flows (i.e., hydrogen, air) controlled by on-board electronic pressure controllers (Parker Hannifan Corporation), and signals recorded using a custom Labview program (National Instruments).

System 1 allowed concurrent analysis of multiple analytes from commercially available mixtures to facilitate analysis of large numbers of analytes but allowed only limited analysis of highly oxygenated compounds due to their inability to elute from a GC column.  System 2 addressed this limitation by permitting injection of aqueous solutions of more oxygenated compounds that could not elute through the GC. System 3 is functionally similar to System 2b but approaches the configuration of a potentially field-deployable desorption and detector train by being more portable and cheaper than the Agilent-based System

2 and allowing direct connection between the passivated cell and the detectors. The metal desorption cell used in Systems 2b and 3 was a passivated steel cell with an attached heater cartridge for temperature control, identical to that previously described for aerosol sampling and desorption using an internal impactor jet (removed for this work) as part of the field-deployable Thermal desorption Aerosol Gas chromatograph (TAG) (Kreisberg et al., 2009).

Sample data from System 1 is shown in Figure 2b. FID response and $CO_2$ response are closely coupled, with a delay in the

$CO_2$ signal of 3-5 seconds due to transit time through the permeation dryer. Resolution on the $CO_2$ channel is slightly degraded due to band broadening during this transit, but chromatographic peaks are nevertheless clear. The resolution provided by these detectors is sufficient for integration of the chromatographic peaks with ~10% uncertainty in most cases (Isaacman-VanWertz et al., 2017) (up to 20% for partially co-eluting peaks like those shown in Figure 2b). Concentrations of $CO_2$ in the flow are only on the order of 1 ppm per 10 ng of analyte, resulting in the relatively low observed signal-to-noise on this channel.

### 2.3 Materials

A variety of mixtures and individual compounds were analysed in this work. To facilitate collecting data on a broad range of atmospherically-relevant compounds and compound classes, commercially available mixtures containing unknown analytes were analysed, including four scents of air freshener and six perfumes and colognes. Analytes in these mixtures were identified by GC-MS. Compounds in these mixtures were analysed only for their $FID/CO_2$ ratios, not to obtain quantitative information on their concentrations. Analytes from these mixtures account for 46 of the total 89 analytes for which $FID/CO_2$ was measured. Dilutions of all fragrances and mixtures were made in methylene chloride (DCM) solvent (1-10% v/v). Aqueous solutions of oxygenated analytes were made with deionized water produced by a Barnstead/Thermolyne NANOpure Analytical Deionization System, Model D4744. Individuals compounds analysed were provided by Sigma-Aldrich, Acros Organics, Fisher Scientific, Alfa Aesar, and Fluka, all at purities of at least 98%.

### 2.4 Experimental details

Calculation of $FID/CO_2$. Systems 1 and 2 provided data for 89 analytes, from which the ratio of FID signal/$CO_2$ signal (FID response per $CO_2$ produced) versus O/C ratio could be plotted. Individual compounds were injected at concentrations of approximately 250 ng/µL, either as aqueous solutions or in a carbonaceous solvent purged prior to thermal desorption as described. $FID/CO_2$ of oxygenated and unsaturated compounds were normalized to the $FID/CO_2$ ratio of an $n$-alkane analysed at nearly the same time. This approach simplifies interpretation by providing a value of 1 unit of FID response per $CO_2$ produced for $n$-alkanes (which have the maximum possible FID response), and a value less than or equal to 1 for any oxygenated compound. This normalization further corrects for any day to day variability in the sensitivity of the FID or drift in the $CO_2$ instrument. In System 1, which involved GC separation, the $FID/CO_2$ of an analyte was normalized to that of the n-alkane with the nearest retention time. In System 2 and 3, individually injected analytes were normalized to injections of dodecane immediately preceding or following analysis. The selection of $n$-alkane for normalization is not critical, as nearly all saturated hydrocarbons were observed to have an $FID/CO_2$ ratio within 1% of the $n$-alkane average. The $FID/CO_2$ ratio is theoretically independent of the mass of analyte introduced since both FID and $CO_2$ scale with analyte quantity. However, to minimize uncertainty, $n$-alkanes for normalization were introduced at concentrations similar to the analyte of interest, which should account for any potential non-linearity in detector response.

FID combustion efficiency. In addition to corroborating results from Systems 1 and 2 in a configuration closer to that that might get used as a detector train in a field-deployable instrument, System 3 was used to confirm complete combustion efficiency of the FID and analyse multi-component mixtures. To assess combustion efficiency, a known mass flow of $CO_2$ gas ($2 \pm 0.2$ sccm of 1% $CO_2$ in balance air) was introduced to the desorption cell in the same location as the injection of analytes to provide a signal-to-mass response factor for the $CO_2$ analyser. As all flows and pressures are controlled electronically, this

flow of known calibrant undergoes the same flow and pressure conditions as any desorbed analyte. Uncertainty in calibrant flow (10%, due to operating the mass flow controller at the low end of its full scale) and $CO_2$ dominates over other sources of uncertainty (e.g. analyte mass injected) in this calibration. Mass of carbon introduced as an analyte was compared to mass of $CO_2$ detected. Four analytes were tested for complete combustion spanning the range of O/C as described in Section 3.

Multi-component mixtures. To confirm that results for individual analytes produced predictable results when combined into simple multi-component mixtures, individual analytes were combined into solution at varying relative concentrations. Components were selected with differing O/C ratios to provide large changes in FID response, but similar vapor pressures, $p^0$, to ensure both analytes were desorbed together and reached the FID and $CO_2$ analyser at approximately the same time. These constraints are relatively limiting, as few sets of commercially available compounds were identified to have very similar (known) vapour pressures, but large differences in chemical structures. The four compounds used to meet these requirements were: dodecane, $p^0$ = 18 Pa, O/C = 0; 1-octanol, $p^0$ = 11 Pa, O/C = 0.125; hydroxyethyl methacrylate, $p^0$ = 17 Pa, O/C = 0.5; and propylene glycol, $p^0$ = 17 Pa, O/C = 0.66 (vapor pressures from EPI Suite software values at 25°C, (US EPA, 2019)). Due to their relatively minor differences O/C (and measured FID/$CO_2$ ratios), mixtures of dodecane and octanol were not considered, nor were mixtures of hydroxyethyl methacrylate and propylene glycol.

Calculation of structure-based ECN. For comparison of measured data to ECN, estimated ECN was calculated based on the criteria of Scanlon (Scanlon and Willis, 1985). ECN for aromatic compounds with multiple functional groups were determined based on different literature (Jorgensen et al., 1990). Relative ECN is calculated as the ratio of this estimated ECN to the number of carbon atoms in the analyte. A more detailed look at calculating rECN and the conversions used are given in Supplementary section S2. Chromatographic peaks and thermograms were analysed and integrated using the publicly available TERN software package (Isaacman-VanWertz et al., 2017) in the Igor Pro programming environment (Wavemetrics, Inc.). For data with replicate measurements, potential outliers were discarded based on Dixon's Q test with a 95% confidence level.

## 3 Results and Discussion

### 3.1 Complete combustion by FID

Quantification of $CO_2$ produced from the analysis of known amounts of analytes provides an estimate of the efficiency of the conversion from organic carbon to $CO_2$ in the FID. Though FIDs are designed for complete combustion, incomplete conversion to $CO_2$ (due to e.g., incorrect hydrogen-to-air ratios) could result in high error or variability in measured FID/$CO_2$ ratios, particularly if combustion efficiency is related to molecular structure. Combustion completeness is investigated in Figure 3, depicting results for four different analytes of varying degrees of oxygenation, along with a 1:1 line for reference. The average conversion of all analytes is 94±9 %, within error of complete combustion. This standard deviation is actually lower than the estimated 15% uncertainty in the amount of $CO_2$ measured (combined 10% uncertainty in calibrant flow and 10% uncertainty

in peak integration); uncertainty in amount of carbon injected is comparatively lower, estimated as 5% uncertainty in solution concentrations and injection volumes. Less oxygenated analytes (squalene and diethyl phthalate, introduced as solutions in DCM) exhibited efficient conversion with highly reproducible results: 95% conversion and a relative standard deviation (RSD) between replicate injections of ~5%. More oxygenated components, which were introduced as aqueous solutions, were more variable. Hydroxyethyl methacrylate ("HEM") had a mean conversion of 100%, but with a somewhat more variable RSD of 13%. Propylene glycol had a mean yield of only 87% and an RSD of 7%. These data are tabulated in Table S6 of Supplementary Information Section S6. These differences may be explained in part by solvent effects. The DCM could be evolved entirely before heating the cell, yielding higher precision for squalene and diethyl phthalate trials. However, solvent blanks of water gave small signals on the $CO_2$ detector and corrections were made to the HEM and propylene glycol peaks. As concentrations of HEM and propylene glycol became more dilute, the background water signal became comparatively large and uncertainty grew. Overall, however, the four compounds showed strong linearity and high percentage yields, supporting the conclusion that the FID converts all analysed carbon to $CO_2$ without strong biases due to molecular structure.

## 3.2 Correlation between measured variables

Figure 4 shows correlations between three parameters for the 89 analytes: FID/$CO_2$ signals, estimated relative ECN, and O/C. FID/$CO_2$ is the measured amount of FID signal generated per $CO_2$ produced, which is, assuming complete conversion of all carbon in an analyte, the amount of FID signal per carbon atom in the analyte. By normalizing this value to an n-alkane, FID/$CO_2$ provides a measure of the amount of FID signal generated per carbon atom in the analyte relative to a hydrocarbon, which is the definition of rECN. This observation suggests that rECN should equal the measured FID/$CO_2$, and they are indeed observed to correlate closely (Figure 4b). FID/$CO_2$ tends to be slightly lower than expected, which is likely due in part to uncertainty in structure-activity based estimation of ECN, which has been previously shown even for hydrocarbons to be on the order of 10% with a tendency to overestimate (Faiola et al., 2012) . Close correlations between FID/$CO_2$ and both rECN and O/C indicate that rECN and O/C must also be correlated, which is shown to be true in Figure 4c. Uncertainty in the average trends of these relationships is very low, with uncertainty in the fitted slopes of less than 4% in all cases (uncertainty in all fit coefficients provided in Table S7 of Supplementary Information Section S7). For 14 of the analytes shown, FID/$CO_2$ was measured in more than one instrument configuration, with results from one configuration always within 7% of the average value for an analyte (Supplementary Information Section S5, Figure S2). FID/$CO_2$ is therefore largely independent of the mechanism by which an analyte was thermally transferred, and uncertainty in the measured FID/$CO_2$ of an individual component is on the order of 15%, in agreement with the more formal analysis of errors discussed in Section 3.3.

Comparison of Figures 1 and 4c shows that the fitted slope of rECN versus O/C falls well within the boundaries demarcated by the carbonyls and carboxyl groups, -1.0 and -0.5, respectively, and that the compounds used in this work follow the same trends as previously published ECN data. The exact slope of this line will depend on the analytes measured, so the

functionalities and vapour pressures of the 89 analytes are provided as Table S5 in Supplementary Information Section S4. Methanol (open symbol in Figure 4) is an apparent outlier and excluded in these fits because it may be attributable to a unique

feature of methanol combustion in an FID. Specifically, during pre-combustion in the hot hydrogen-rich environment of the FID, multicarbon alcohols lose water through elimination to form an alkene, a pathway that is significant for larger alcohols but is not available to methanol (Holm, 1997). We speculate therefore that methanol falls off the line as it undergoes a fundamentally different combustion process than all other alcohols, so its behaviour as an outlier does not have negative implications for the application of this system to larger compounds. Furthermore, any analytes larger than $C_1$ will be less

influenced by the anomalous behaviour of one functional group, so any similar outlying behaviour would be less substantial (i.e., other carbon in the molecule would counterbalance the effect). Overall, the observed FID/$CO_2$ ratios of mixtures correspond quite well with the expected ratios across the full range of anticipated measurements, demonstrating that the FID/$CO_2$ ratios observed for single components are maintained in mixtures.

The relationships shown in Figure 4 are useful for applications in which analytes are pre-separated (e.g., chromatographic

instruments). However, in order to be useful in bulk analyses of atmospheric particles or other mixtures, a potentially useful application, the observed relationships between O/C and other parameters must hold for mixtures as well as individual analytes. The principles of this measurement approach suggest that a mixture should respond as the weighted contribution of its constituents, since all carbon entering the FID was shown to combust to $CO_2$ (Figure 3), and no previous literature indicates that the co-detection of two components biases the FID response to one of those components. To demonstrate a lack of any

specific bias in FID (or $CO_2$) response due to co-detection of multiple analytes, mixtures were analysed comprised of varying fractions of two components. Results from three separate mixtures containing two components at a time are shown in Figure 5. The expected FID/$CO_2$ was calculated as the average of the two pure components weighted by their carbon fraction in the mixture. The measured FID/$CO_2$ was the experimental value of the mixture. An orthogonal linear fit of the three tested mixtures (each at seven varying relative fractions of each component) shows a slope of 0.95, within uncertainty of unity. Experimental

mixtures in this work were limited to two components due to the need for comparable vapour pressures but differing instrument responses. However, extrapolation to more complex mixtures is supported by both the theoretical principles of the approach, and previous work on measurement of O/C by the Aerosol Mass Spectrometer (AMS). That instrument is similarly calibrated as simply the average relationship between analyte O/C and the measured parameter (for the AMS, O/C of molecular fragments) for a large number of individual analytes (Aiken et al., 2008; Canagaratna et al., 2015). Canagaratna et al. found

that uncertainty is actually highest in applying the average relationship to one or two components, and decreases with mixture complexity as the average relationship better describes the complex mixture (Canagaratna et al., 2015). The relationship observed between FID/$CO_2$ and analyte O/C can therefore be expected to extend to more complex mixtures, though application of this relationship for bulk measurements of real-world ambient aerosols would need to first be validated.

## 3.3 Error estimates

The major benefit of quantifying the relationships between O/C, rECN, and measured $FID/CO_2$ is their potential use in predicting one parameter from another. To understand the error in such a prediction, it is first useful to evaluate how precisely any of these parameters are known. O/C is known precisely for each analyte, and published uncertainties in ECN are on the order of 10% (Faiola et al., 2012). Uncertainty in $FID/CO_2$ is, practically speaking, dominated by uncertainty in the data analysis. All flow rates and pressures are controlled and known to within 2%, and precision of the $CO_2$ detector is similarly negligibly small. Because $FID/CO_2$ is theoretically concentration independent, uncertainties in detector calibration and injection volumes are unimportant, and are low in any case (10% and ~5% respectively). FID and $CO_2$ data can therefore be generated precisely, and uncertainty is primarily driven by analysis, specifically the estimated 10% uncertainty in peak integration (Isaacman-VanWertz et al., 2017). Calculating $FID/CO_2$ requires integrating two different peaks, so combined uncertainty in this parameter is ~15%.

To assess the accuracy in calculating unknown variables from observed values for individual analytes, absolute and relative (%) errors are shown in Figure 6. Error in predicted values is less than 20% in nearly all cases, suggesting error in this relationship is not dominated by uncertainty in parameters themselves, but rather inherent error is applying the average relationships to individual analytes. However, error is often much lower and exhibits some heteroscedasticity worth discussing. Generally, absolute error is higher for oxygenated components due to the structurally-dependent effects of oxygen, which are ignored in these relationships (e.g., the divergence of the carbonyl and carboxyl trends). However, prediction of O/C from $FID/CO_2$ at low O/C yields very high relative error despite low absolute error because as O/C approaches 0, even low absolute errors imply high relative error. For highly oxygenated compounds, relative error in O/C appears to plateau to about 20%; this level of uncertainty for individual analytes is comparable to that of the AMS. Error in estimating O/C from $FID/CO_2$ for an individual analyte can therefore be reasonably summarized as a relative error of 20% with a minimum absolute error of approximately 0.05. Generally, the rECN can be predicted from either $FID/CO_2$ (Figure 6b) or O/C (Figure 6c) with errors of 10-15% for highly oxygenated compounds (consistent with propagated error estimates), and <5% for less oxygenated compounds (O/C < ~0.5). These low errors indicate that the sensitivity of an FID can be estimated with high certainty either directly from O/C or, in the absence of this information, from a direct measurement of $FID/CO_2$.

We further consider here the expected error in the application of these relationships to mixtures, as opposed to individual analytes. The average trends in these relationships have low uncertainty (<5% uncertainty in slopes), so assuming that a sufficiently complex mixture approximates the central tendencies of the relationships, uncertainty in the application of these relationships to more complex mixtures will be dominated by the ~15% uncertainty in the parameters themselves. As described in Section 3.2, this assumption is supported by previous work by Canagaratna et al. that demonstrated a reduction in uncertainty in moving from single analytes to complex mixtures (Aiken et al., 2008; Canagaratna et al., 2015), with error for a complex

mixture 2-3 times lower than error for a single analyte. Given that error in these relationships for individual analytes is on the order of 20%, error for complex mixtures would again be expected to be dominated by uncertainty in the parameters themselves. In other words, overall uncertainty in the application of these relationships to complex mixtures is expected to be on the order of 15%. As discussed in the section below, for some potential applications of these relationships to atmospheric conditions, there are additional considerations that could lead to biases not able to be captured by the formal analysis of error here.

## 3.4 Extension to atmospheric samples

This study examines the relationships between O/C, relative ECN and FID/$CO_2$, which have a number of possible applications for use in atmospheric instrumentation. Such applications may use the relationships described in this work to estimate parameters of an individual analyte or may seek to apply them to bulk mixtures such as organic aerosol. We discuss here additional atmospheric issues that might need to be considered, depending on the specifics of the application. In particular, we identify two major sources of potential uncertainty in applying these relationships to atmospheric samples: (1) biases in the set of analytes used to build the relationships shown, and (2) presence of atmospheric constituents such as heteroatoms that may bias the detector signal.

The first issue recognizes that the relationships derived are quantitatively described by the weighted average of the set of analytes investigated. If a compound or mixture analysed is not well described by this set of analytes, it may bias application of the average relationship. For instance, some major atmospheric constituents are not expected to obey the average relationships (e.g., glyoxal, O/C = 1.0, rECN = 0), so, as discussed, error is expected in the application of these relationships to individual analytes. However, for individual components, error is on average 10-20%, and in cases where high accuracy is needed for only one or two components, the relationships here are probably not the preferred approach. For complex mixtures, such high-error compounds will only introduce error to the extent that they are also major contributors to the mixture; in most cases, atmospheric samples are sufficiently complex that any error for one component would not introduce significant error for a mixture. Consequently, the main source of potential bias for atmospheric applications is not error in any specific subset of components, but in whether or not the overall derived relationship accurately describes the mixture to which it is being applied. Confirmation of the appropriateness of the derived equations to any given atmospheric mixture, such as bulk ambient organic aerosol, would require comparing any instrument relying on these relationships to established instrumentation, such as an AMS. We note that the O/C calibration for the AMS is largely based on the same approach used here of developing an average relationship from data for many atmospherically relevant individual analytes and that error of this approach tends to decrease with increasing complexity (Canagaratna et al., 2015), so there is precedent for the successful adoption of this approach. Furthermore, analysis of ambient aerosols has indicated that on average atmospheric oxidation adds approximately equal parts double-bonded (e.g., carbonyl) and single-bonded oxygen (e.g., hydroxyl), which would be expected to yield an expected average relationship between FID/$CO_2$ and O/C reasonably similar to that measured here.

The second issue acknowledges the potential uncertainties caused by the presence of nitrogen- and sulphur-containing compounds in atmospheric samples. In some applications of these relationships, such issues could potentially be avoided, for instance by limiting analyses to low-$NO_x$ environments, or screening out components that have nitrogen in their molecular formula (e.g., if using this approach to calibrate a mass spectrometer, Application #2 below). However, clearly some applications of these relationships (e.g., bulk analysis or organic aerosol, Application #1 below) require a thoughtful treatment of heteroatoms, so we discuss the issue in some detail here. The effects of nitrogen and sulphur on FID response are less certain than oxygen-containing functionalities. Reduced nitrogen (specifically amines) has been shown to impact FID response similar to alcohols (i.e., single-bonded oxygen), so any amines (and likely other C-N bonds) present in a sample would produce an effect that appeared to be caused by oxygen. Conversely, though nitrate (-$RONO_2$) groups do not have well-studied FID responses, it is reasonable to expect that the C-O bond may have the impact of the C-O bond of an alcohol, and the other heteroatoms (one nitrogen and two oxygens) would likely have no effect on the signal as they are not bonded to a carbon atom (2010). Such a scenario would imply that nitrate groups would bear three oxygens, but cause the effect of only one oxygen, in essence masking two oxygens. These expected impacts suggest that each nitrogen produces signal of between +1 oxygen and -2 oxygen. Sulphur would likely exhibit similar effects. As with other sources of error, these effects may yield high uncertainty for individual components, but for bulk mixtures will be proportional to the amount of nitrogen present, and will likely be mitigated by the presence of a mixture of reduced and oxidized heteroatom-containing groups. Given nitrogen and sulphur are present in atmospheric mixtures at concentrations around ten times lower than oxygen, this would yield only a 10-20% error, which is not substantially beyond the overall uncertainty in the relationships. The specific effect of nitrates "masking" two oxygens was actually previously shown to impact the AMS as well and produce 10-20% error (Farmer et al., 2010), consistent with the analysis here. One nitrogen-containing compound (musk ketone) was studied in this work and did not exhibit significant bias but was excluded from analysis as the relevant structure-activity relationships do not include estimated impacts of the substituent nitro groups. Overall, the complication of heteroatoms in atmospheric applications of these relationships is highly dependent on the application. If used to calibrate individual components that are likely to have high nitrogen content, nitrogen will introduce high uncertainty. However, if applied to bulk ambient aerosols, the error introduced by heteroatoms is on the same scale as existing uncertainty in these relationships. Any user of these relationships should then consider the potential impact of heteroatoms in their specific application. In applications for which heteroatoms need to be accounted for explicitly, the ability to run this detector chain in parallel to other common GC detectors (e.g., a flame photometric detector for sulphur) may allow improved understanding of heteroatom effects.

450

**4 Conclusions and Applications**

This work demonstrates that the carbon and oxygen content of single compounds and mixtures can be directly measured by coupling an FID and a downstream $CO_2$ analyser. Specifically, three major conclusions support this claim:

1. Complete combustion (within uncertainty) in an FID of a wide range of organic compounds allows direct quantification of analysed carbon as the amount of $CO_2$ produced

2. Oxygen content (as O/C) is closely correlated with the amount of FID signal produced per $CO_2$ generated (FID/$CO_2$), as well as existing structure-activity estimates of per-carbon FID sensitivity (rECN). These correlations extend to multi-component mixtures.

3. Uncertainties in these parameters and relationships between them is ~15%, and in the prediction of an unknown parameter from a known parameter for an individual analyte is typically <20%.

The correlations between O/C, FID/$CO_2$, and rECN quantified in this work may advance the field of atmospheric chemistry through a variety of possible applications we consider here:

Application 1. The coupled detector train described in this work could be coupled with a sampling and thermal desorption system as a field-deployable instrument for measuring of carbon and oxygen content of particles based on the relationship between FID/$CO_2$ and O/C. These detectors are more robust and lower maintenance than currently available instrumentation for the automated characterization of aerosol chemical composition, though would also provide lower chemical detail compared to mass spectrometric instrumentation. However, it could provide O/C, an important parameter for aerosol chemical modelling and understanding aerosol impacts. Moreover, the use of GC detectors as an instrument platform allows potential inclusion to additional of other detectors (e.g. FPD for sulphur or phosphorus, NPD for nitrogen or phosphorus) permitting a more comprehensive view of the chemical composition of aerosols, and correction for some uncertainties caused by the presence of heteroatoms. This application would require overcoming additional technical hurdles in sampling and thermal desorption, but approaches have been previously demonstrated for online sampling and thermal desorption of particles (Kreisberg et al., 2009; Williams et al., 2006; Zhao et al., 2013). Furthermore, as discussed in Section 3.4, the compounds used to develop the relationships in this work may not reflect the average composition of ambient aerosols, which may contain functional groups not represented here (e.g., peroxides, nitrates, etc.), so application of the demonstrated relationships in an atmospheric context would consequently require some comparison with currently accepted approaches to measure O/C.

Application 2. An FID could be used as a calibration tool for new instrumentation without requiring molecular structural information to estimate FID response. While the FID is an attractive near-universal detector, the structural dependence of its response has limited its adoption. However, we demonstrate here that FID sensitivity can be robustly estimated with low uncertainty from O/C or measured FID/$CO_2$ ratio. This implies that any instrument that can be coupled to an FID can use it for quantification, regardless of whether a molecular formula is available. For

instruments that do provide a molecular formula, quantification by FID is possible even without including the additional complexity of a $CO_2$ analyser. For example, by applying the relationship between O/C and rECN, an FID in parallel with a chemical ionization mass spectrometer could use the molecular formula from the mass spectrometer to estimate FID sensitivity. This could allow improved understanding of response or sensitivity of new atmospheric measurement approaches.

Application 3. A $CO_2$ analyser could provide an additional dimension of chemical resolution for an FID being used as a GC detector. For example, identification of an analyte by its retention time could be confirmed by its FID/$CO_2$ ratio.

These possible applications provide a demonstration of the utility of the novel approach presented here. Quantifying the average relationships between FID sensitivity, O/C, and a directly measurable parameter opens the door to a wide range of potential new moderate cost measurement techniques that may find use in the field.

*Data availability.* Measured and actual parameters are provided as Supplementary Data in Excel format. Additional data available upon request.

*Author contributions.* NMK, BS, SVH and PK developed the hardware and conceptualized the theory of operation of the instrumentation. CB and PK contributed to data collection. JFH was in charge of data collection and writing the manuscript, working under the direction 0f GIVW.

*Competing Interests*. The authors declare that they have no conflicts of interest.

*Acknowledgements/Financial Support*. This work was supported through the Department of Energy SBIR/STTR Program, grant number DE-SC0018462.

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

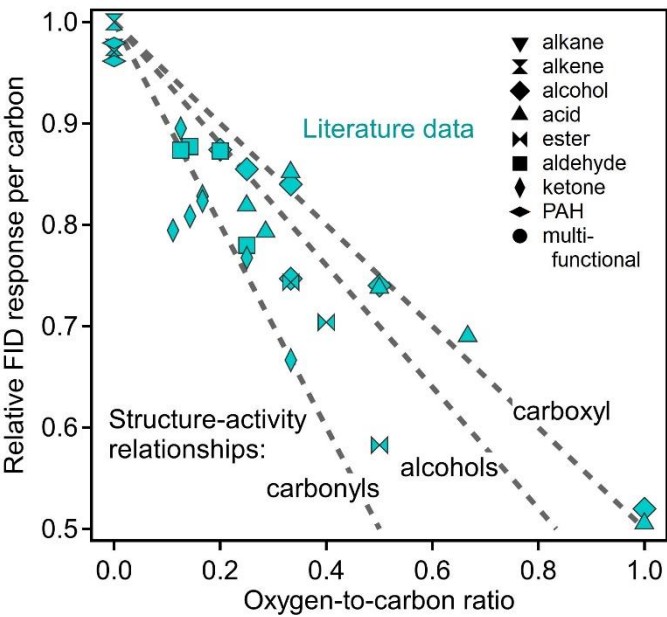

**Figure 1: Plot of relative ECN versus O/C ratios based on literature values. Literature data are from Scanlon and Willis (1985), representing a variety of functional groups shown as different shapes. Dashed lines are the theoretical slopes of compounds comprised completely of labelled functional groups, based on the structure-activity relationship provided by Scanlon and Willis (1985). In contrast to that of the carbonyls and carboxyls, the theoretical slope of alcohols is not structure independent and is based on a chosen subset of alcohols. Compounds used to develop the alcohol slope are provided in Table S4.**

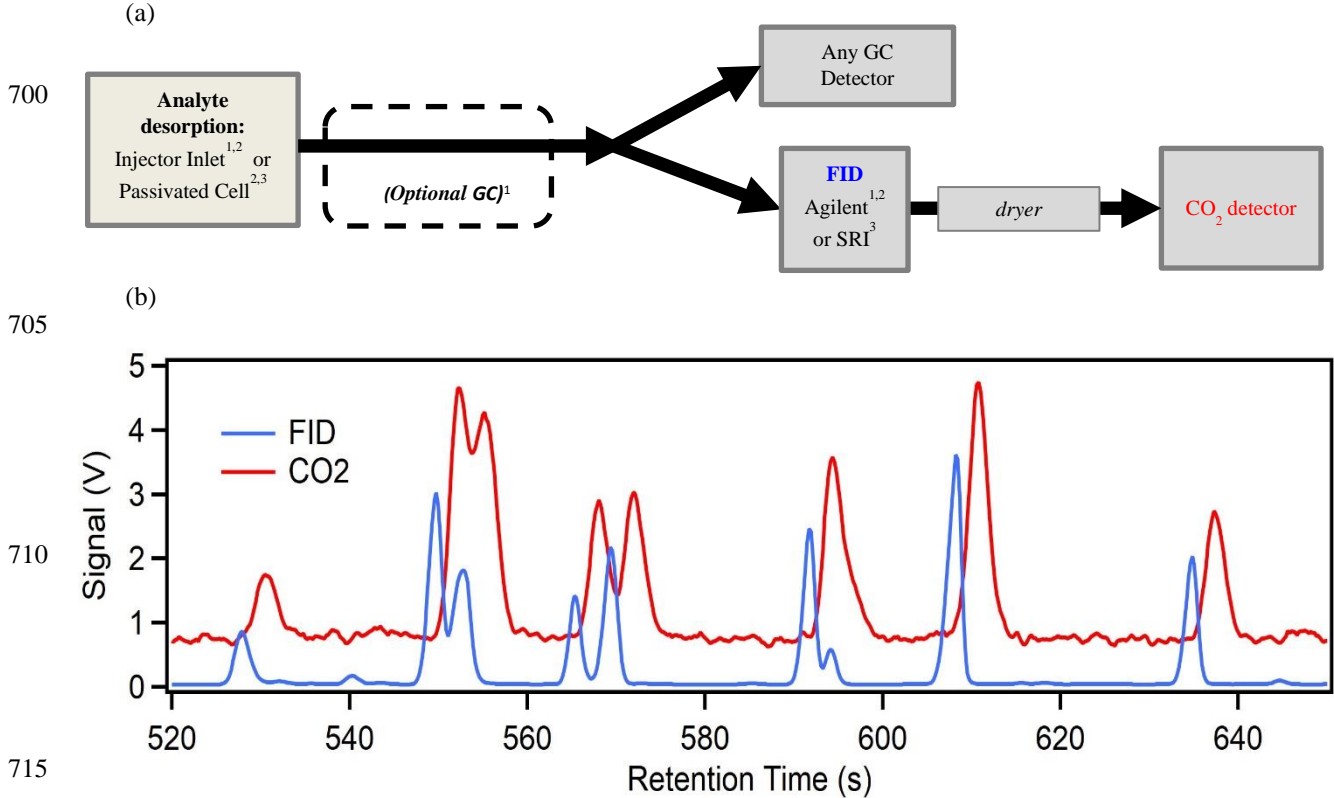

**1,2,3 Superscripts indicate configurations for different systems describe in Section 2.2**

**Figure 2: (a) A generalized schematic of the instrument configurations (b) Sample data of simultaneously measured FID and $CO_2$ signals collected on System 1 (i.e., including optional GC)**

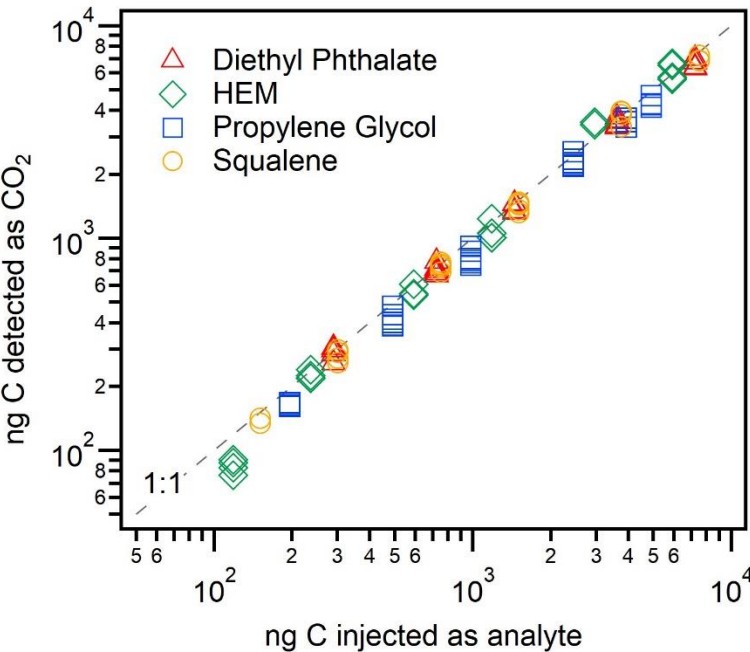

**Figure 3: FID combustion efficiency, shown as ng of carbon measured by the $CO_2$ detector versus ng C injected as one of four analytes: squalene ($C_{30}H_{50}$), diethyl phthalate ($C_{12}H_{14}O_4$), hydroxyethyl methacrylate ($C_6H_{10}O_3$) and propylene glycol ($C_3H_8O_2$). Uncertainty in the y-axis is approximately 15% and in the x-axis is <10%; error bars not included for clarity. Percent conversion for each compound and overall is tabulated in Supplementary Section S6.**

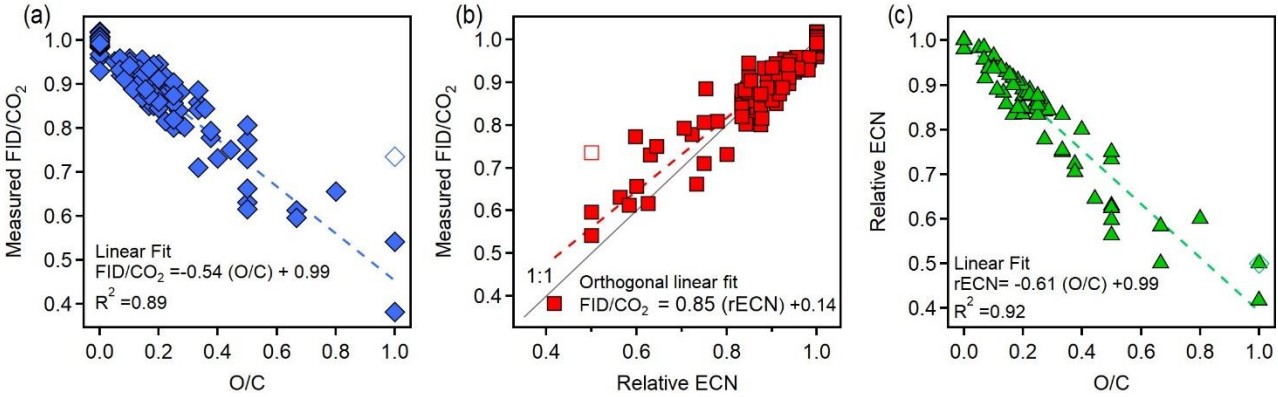

**Figure 4: Plots relating the three variables: measured FID/$CO_2$ relative to *n*-alkanes, relative ECN, and O/C. Comparisons shown
are (a) FID/$CO_2$ versus O/C, (b) FID/$CO_2$ versus relative ECN, and (c) relative ECN versus O/C. Dashed lines are linear fits; fits assume error only in dependent variable in the case of comparisons to O/C (which has no error), and assume error in both variables ("orthogonal fit") in the case of rECN comparison to FID/$CO_2$. Methanol is shown in each plot as an unfilled marker as there are physical reasons it may be an outlier (discussed in the main text). The respective percentage error of the slope and intercept, respectively, for each relationship are: (a) 4%, 1 % ; (b) 4 %, 22%; (c) 3%, 1%.**

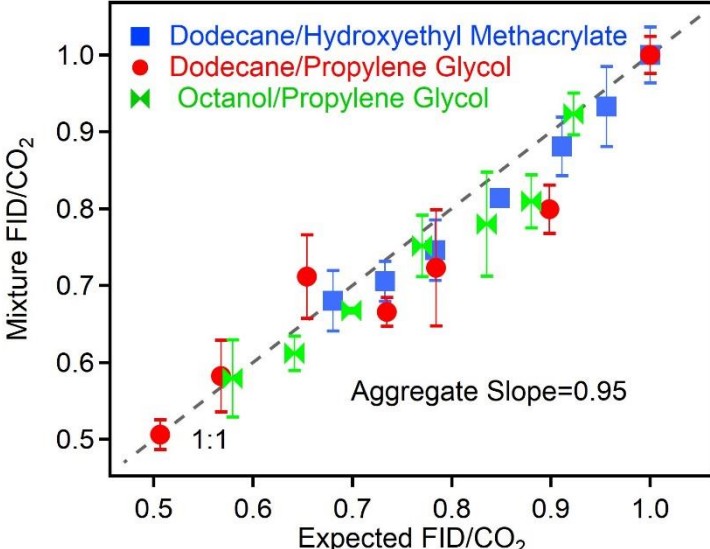

**Figure 5: Measured FID/CO₂ of mixtures compare to the expected FID/CO₂ based on the weighted carbon fraction of the individual components. Each point represents the average of 3-5 replicates, and error bars show standard deviations, which are less than 10% in all cases. Dashed line shows 1:1 line.**

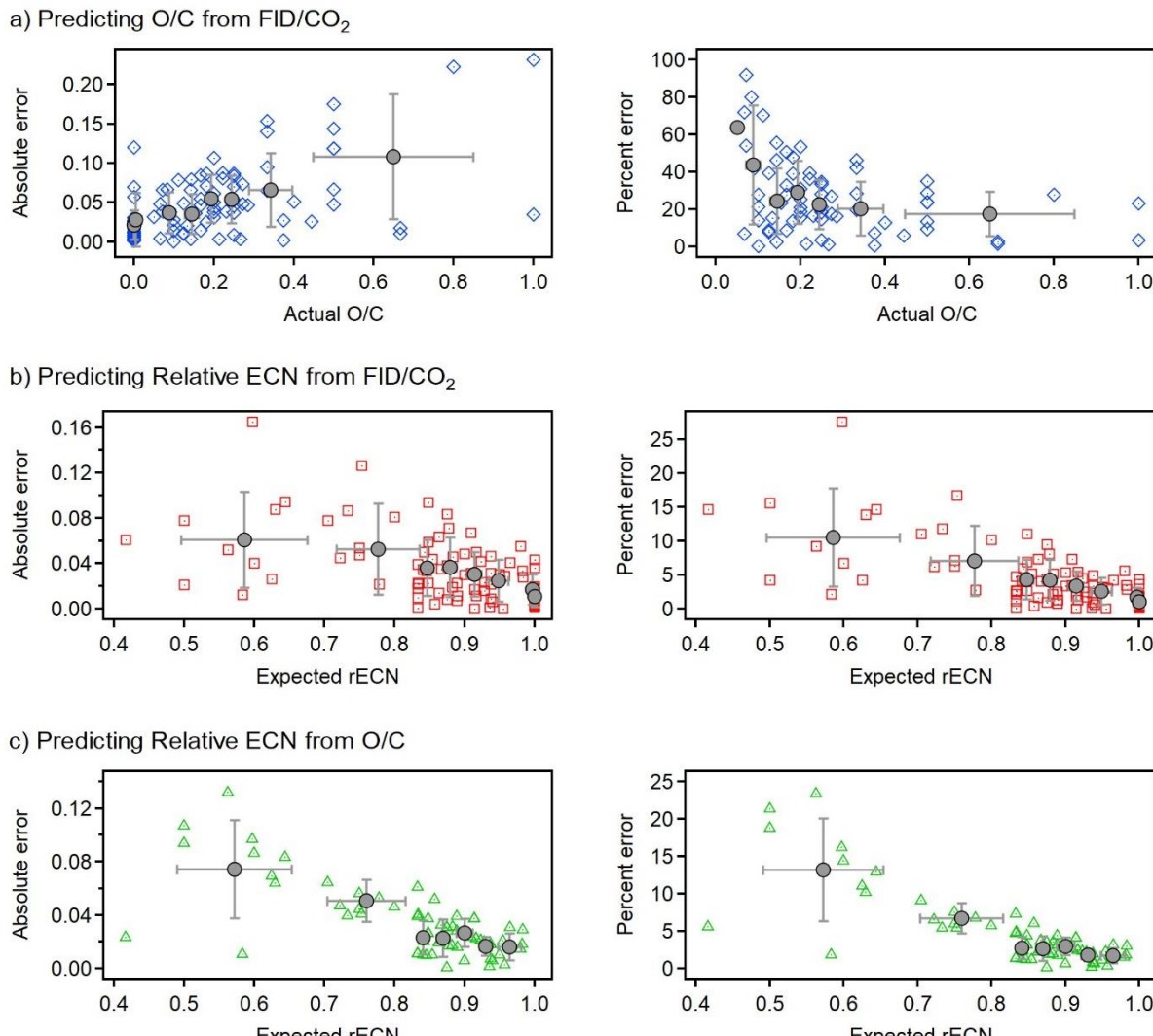

**Figure 6: Absolute error (left panels) and relative error (right panels) in the prediction of (a) O/C from FID/CO₂, (b) rECN from FID/CO₂ (c) rECN from O/C. All errors shown against the actual value of the predicted value (i.e., compound O/C, or rECN estimated from structure-activity relationships). Relative error calculated as |observed-actual|/actual*100%. All analysed compounds (N=89) shown as individual points with the same colours and shapes as Figure 3. To make trends more qualitatively clear quantiles of equal number of points are shown in grey (quantile average with standard deviation as error bar).**
