# Peer review of "A new approach for measuring the carbon and oxygen content of atmospherically-relevant compounds and mixtures"

_Atmospheric Measurement Techniques, 2020_

## Referee Comment (RC1) · Anonymous Referee #1 · 13 Apr 2020

General Comments:

As is indicated by the title of the paper, a unique, lower cost method to estimate analyte O/C in real time by using FID and a carbon dioxide detector is presented. The importance of understanding ambient O/C is successfully established. Increased oxygen content of aerosol can influence volatility and hygroscopisity and thereby impact climate. Additionally, toxics in the atmosphere, which impact human health, are often the result of oxidation of biogenic precursors. Several higher cost continuous and non-continuous methods of measuring O/C are discussed. This work suggests coupling a flame ionization detector to a carbon dioxide detector for real time estimation of O/C.

[Figure]

The general apparatus schematic is clear and simple, however, it may be helpful to point out the three different "systems" on the apparatus figure 2a. Known concentrations of single compounds (with variable O/C) were injected through the apparatus in order to establish that: 1) FID/CO2 correlates with O/C; 2) FID/CO2 correlates with relative effective carbon number; and thus 3), O/C and relative effective carbon number are also correlated. Simple mixtures of 2 organic compounds were analyzed using a GC coupled to the FID+CO2 detector apparatus. The expected FID/CO2 trends nicely with measured FID/CO2. The authors argue that this trend indicates this apparatus can be used in the analysis of atmospheric particles.

The overall presentation is clear and concise. Appropriate high-quality references are made. Results suggest that this apparatus could be successful in the field for O/C, however, additional, more complex testing would make a more convincing case (See Specific Comments #3 and 4). Given the scope of this paper, I do not believe that additional testing is necessary prior to publications. I would recommend further testing prior to publication of field data (either in the lab using more atmospherically relevant conditions/mixtures or comparison tests along side a well established instrument that measures O/C).

Specific Comments:

1) In the "Materials" section, complex commercially available mixtures on unknown compounds (perfumes, colognes, etc.) are mentioned. I do not recall any discussion regarding the analysis of these complex unknown mixtures. Did I miss something? If I did not miss something, I am unclear why they are being discussed in the materials section.

2) As I read this paper, my initial thoughts went immediately to, "This looks great in the lab, but are the concentrations you used in the lab relevant to the field? Can this apparatus handle the concentrations you would expect to be measuring in the field?" The SI covers this nicely, but it may be an important to note or reference in the main

manuscript as well.

3) It is not clear to me how this apparatus would function in the field. If you are interested in the O/C of atmospheric particulate, how do you plan to deal with the presence of organic gasses entering your apparatus?

4) While I recognize the difficulty in replicating atmospheric conditions in the lab, I have a hard time accepting that a mixture of 2 compounds indicates that this method of O/C estimation would be successful in the field. Did you think about testing more complex mixtures? Even a mixture of 2 compounds would provide more adequate evidence that this method would work in the field.

5) Methanol was pointed out as an outlier. Are you confident that there are not many more atmospherically relavent "outliers"? Could this potentially skew the trends you are seeing and lead to poor estimations of O/C?

6) Is there a reason no nitrogen- or sulfur-containing organics were tested? Would you expect any changes in the $FID/CO_2$ trends in areas rich with these compounds?

Technical Corrections:

1) There is no mention of where the data in Figure 4 (section 3.2) came from. I assume it is from the 90 different compounds that are mentioned in section 2.4.

2) Line 35: appears to be space between parentheses and period.

3) Line 71: two commas after "Generally"
* * *

---

## Referee Comment (RC2) · Anonymous Referee #2 · 6 May 2020

This work describes the design and lab testing of a lower-cost and simplified method to determine the carbon and oxygen content of particulate matter. In this manuscript, the authors couple a NDIR $CO_2$ detector to a flame ionization detector to demonstrate a new approach to measure the carbon and oxygen content of atmospherically-relevant organic compounds. Three criteria need are tested through theory and fundamental validation of the approach: 1) FID combustion efficiency 2) FID response per carbon atom (i.e., measured ratio of signals, FID/$CO_2$ ) must be inversely proportional to the oxygen content of analytes and mixtures, 3) achieve high correlation amongst FID/$CO_2$ signal ratios, analyte O/C, and analyte FID sensitivity (rECN) within reasonable error. Field testing is not a component of this first method paper. With some minor improve-

ments, I consider this work worthy of publication.

Here are some specific comments and questions.

1. Line 71: eliminate one of the commas after "Generally"

2. Supplement: would be helpful to also list the ECN structure-activity relationship values from the Scanlon & Willis, and Jorgensen publications.

3. Figure 1: The caption should be changed to reflect that dashed line for alcohols represents the rECN calculated for a subset of alcohols, and is not structure independent like with carbonyls and carboxyls.

4. Paragraph starting at line 135: when utilizing a GC/MS system to assist in identification and relating to FID through retention time indices, were there differences in flow rate settings (constant pressure vs constant flow) utilized between the measurement methods, and if so, how was retention time matched/correlated?

5. Lines 141-142: Please clarify how closely the retention index of a given compound needed to match the NIST values.

6. Paragraph starting at line 135: It would help the reader if a more detailed instrument diagram for systems 1,2(a/b),3 were provided in supplemental information.

7. What is the complete list of supporting compressed gases required for this instrument? Helium, Hydrogen, (Nitrogen for FID?), compressed air for FID?

8. Lines 179-183: it's stated the FID/CO2 ratios measured with systems 1&2 were normalized by the ratio of the nearest-eluting alkane. However, with system 2 there was no temperature ramp. Please clarify how the normalization for samples measured by this setup occurred.

9. Figure 4: it would be helpful to have a table of compounds used in these figures within the supplement. What is the volatility range tested, range of functional groups, or some multifunctional compounds too?

10. Lines 236-237: The sentence "It is notable but probably incidental that the observed downward slopes and R2 values match closely with those of a set of alcohols, -0.58." seems ambiguously worded, and relies too heavily on the reader to have read section S2 in the supplement to understand what is meant.

11. Paragraph starting at line 243 and Figure 5: why did the authors select these specific mixtures?

12. Line 269: why would we want to determine rECN when we already have O:C from measured FID/CO2, which would be the info utilized by atmospheric scientists.

13. The technique shows good promise, it will be interesting to see how it performs on ambient mixtures in the field, relative to the existing high-cost, high-maintenance techniques.
* * *

---

## Referee Comment (RC3) · Anonymous Referee #3 · 8 May 2020

I am puzzled by the disconnect between what the paper says is a main goal – to develop a fast/easy/inexpensive way to measure organic carbon and O/C in particulate matter, and the fact that the authors present no measurements of particulate samples. The paper does describe a method for measuring the carbon and oxygen content of organic compounds that could potentially be used for particulate samples, but how that application would work is not explained and certainly not demonstrated. The method involves a combination of flame ionization (FID) and carbon dioxide ($CO_2$) detection schemes, and the ratio of FID to $CO_2$ response is used to estimate the O/C ratio of organic species. The paper describes a potentially interesting and useful technique, but there are major issues with the paper, and it will need considerable work before it

is acceptable for publication. I have the following general and specific comments that need to be dealt with.

General Comments:

No particle-phase, or particulate-like samples were analyzed in this work. If the authors were to try to emulate particle organic carbon (POC) they would realize they should also be working with compounds like oxalic acid and glyoxal, which have O/C ratios of 2 and 1, but probably no FID response. There is potentially a lot of oxalic acid/oxalate – which would represent a large deviation from the clusters of points that are presented by the compounds studied in this work. There needs to be some discussion of how such compounds would affect what is actually observed for POC. In addition, organic nitrates and organo-sulfates should also be explored if the authors really want to demonstrate that this technique works for POC. The only hetero-atom (i.e. non-O atom) containing species, Musk ketone (apparently a di-nitro compound) was excluded from the analysis – because it did not conform. This does not bode well for the technique. What was the nature of the problem that led to this disqualification, and how does that bear on whether the technique can work for nitro-aromatics for example?

The LI-COR instruments used for $CO_2$ measurements have distinct characteristics: the LI-COR 6262 is an older model and has an inherently non-linear response to $CO_2$ that must be calibrated throughout the useful range with multi-point calibrations, and the LI-COR 7000 has a linearized response. The specifics of how these two different instruments were calibrated need to be presented. In addition, these detectors are concentration-sensitive and therefore the systems, at least as far as I can understand them from the diagram, require an accurate knowledge of the flow rates and pressures in the systems. The details of how these were determined need to be presented in this work.

Inherent uncertainty in O/C is simply an unavoidable feature of the technique, but we

don't have any substantive discussion that places this in context. How do the uncertainties in O/C ratios from this technique compare to uncertainties from AMS measurements of O/C? Does this technique really represent an advance in this area or is its' main feature that it is less expensive and easier to field?

Specific Comments:

Page 4, Lines 119-120. This statement needs much more context given that the slopes of rECN to O/C can vary between -1 and -0.5. Is this really good enough for the purposes of what we'd like to do with POC data?

Page 5. The instrument diagrams and associated descriptions are not nearly adequate to explain what was done, where uncertainties could be introduced, and how POC sampling and analysis would work. Doesn't the TAG cell have a filter in it? Was that part of the configuration, if not, how would that change how the system would operate?

Page 6. Lines 165-166, and Figure 2b. It seems clear that there was incomplete separation in the GC peaks, especially for the CO2 detector. How were these analyzed and how did that impact the results e.g. accuracy and precision?

Page 6, Line 183. Doesn't normalizing to the nearest n-alkane create problems and uncertainties? It seems like some oxygenates might be in a completely different retention time range than the number of carbons it has. The carbon count could easily be off by one or more carbons.

Page 7. Section 3.1 I have a hard time believing that FIDs are not 100% efficient in converting carbon compounds to CO2: isn't this known? There are errors in both quantities plotted in Figure 3. Those need to be shown for the data points and other possible sources of systematic error should be discussed. For example, the CO2 instruments are concentration sensitive, so flow rates and pressures need to be known accurately and/or controlled so that they don't change as the experiments are being conducted.

Page 7 Line 224. I had to read this several times to decide I basically disagree with this

statement. Figure 1 suggests there will be considerable uncertainty in the relationship between rECN and O/C but is doesn't imply anything about FID/CO2. The extension to FID/CO2 is really because both carbonyl and carboxyl carbons have no FID response.

Page 8., Line 229. I this $\pm 4\%$ from the fit to the data?

Page 8., Line 258- Page 9, Line 259. This statement doesn't seem correct, especially when one considers compounds like oxalates, glyoxal, glycolic acid and the like. There is an inherent $\pm 25\%$ built into the technique.

Page 11. The Chow et al., reference has an error. Page 11., Line 336. Should be "Association" Page 11., Line 344. Should this be 'GC'? Page 14., Line 442. Correct the typo.

Figure 4 – Please give the uncertainties in the fit parameters, i.e. slope, intercept

Conclusions; The authors have failed to do the necessary work to demonstrate that this technique works for analyzing POC. The whole paper needs to be re-cast to focus on gas phase species, or a considerable amount of additional work needs to be done on actual particle samples and the species that we already know they contain: N- and S- containing organic compounds. As far as the built-in $\pm 25\%$ uncertainty in O/C, the authors need to make a case that their measurement can still be useful in spite of this feature.

---

## Author Comment (AC1) · 20 Jun 2020

We thank the reviewers for their careful reading of our work and appreciate their assessment that this work is interesting. We appreciate the concerns raised by the reviewers, and hope they agree that our revisions of this manuscript satisfactorily address those concerns. Because all three Reviewers have raised similar scientific concerns and discussion points, we have responded to all three Reviewers in a single document. Responses to this Reviewer (Anonymous Reviewer #1) are on pages 2-10.

Please also note the supplement to this comment:

[Figure]

https://www.atmos-meas-tech-discuss.net/amt-2020-44/amt-2020-44-AC1-supplement.pdf

---

## Author Comment (AC2) · 20 Jun 2020

We thank the reviewers for their careful reading of our work and appreciate their assessment that this work is interesting. We appreciate the concerns raised by the reviewers, and hope they agree that our revisions of this manuscript satisfactorily address those concerns. Because all three Reviewers have raised similar scientific concerns and discussion points, we have responded to all three Reviewers in a single document. Responses to this Reviewer (Anonymous Referee #2) are on pages 11-17.

Please also note the supplement to this comment:

[Figure]

https://www.atmos-meas-tech-discuss.net/amt-2020-44/amt-2020-44-AC2-supplement.pdf

**Supplement:**

**Response to Referees**

All reviewers highlighted the specific scope of this manuscript, which is focused on understanding the fundamental relationships between O/C, FID response, and empirical FID/$CO_2$ ratios. These relationships may find use in a range of different atmospherically-relevant applications (such as those discussed in the Applications section, e.g, measurement of bulk O/C, calibration of novel mass spectrometers), and the specifics of any application would dictate the uncertainties or limitations of the approach. We have revised the Introduction of the manuscript to make more clear the specific scope of this manuscript, and note ways these relationships may be applied beyond simply bulk measurements of O/C. As noted by all reviewers, we agree that the development and use of any such instrument relying on these relationships would require additional testing and comparison to existing approaches, as well as demonstrating evidence of overcoming additional technical hurdles. As discussed in the Introduction and Applications sections and noted by the reviewers, a clear driver of the development of these relationships is their application for measuring bulk aerosol O/C. This group is in the process of using these relationships in the development of a field-deployable instrument for the measurement of O/C, and we agree that such an application will require addressing issues as sampling collection, presence of heteroatoms, and comparison to existing instrumentation. These are all issues beyond the scope of the current manuscript, which is restricted to the relationships between relative ECN, FID/$CO_2$ and O/C ratios. We hope the reviewers agree that the revised manuscript clarifies the scope of this work. We have further added a discussion to the Results section, "Extension to atmospheric samples ", which addresses some of the potential complications and uncertainties in trying to apply these relationships to atmospheric mixtures.

The other major comment raised by all three reviewers is on the subject of error and comparison to existing approaches. In this revision, we include several more detailed discussions of these issues, and we provide the reviewers with a detailed comparison to the AMS (which is shown to have comparable error to this approach).

Please find below responses to all individual comments. Reviewers comments are in blue, our response is in black, and any new text is *italicized.*

**Anonymous Referee #1**

General Comments: As is indicated by the title of the paper, a unique, lower cost method to estimate analyte O/C in real time by using FID and a carbon dioxide detector is presented. The importance of understanding ambient O/C is successfully established. Increased oxygen content of aerosol can influence volatility and hygroscopisity and thereby impact climate. Additionally, toxics in the atmosphere, which impact human health, are often the result of oxidation of biogenic precursors. Several higher cost continuous and noncontinuous methods of measuring O/C are discussed. This work suggests coupling a flame ionization detector to a carbon dioxide detector for real time estimation of O/C.

The general apparatus schematic is clear and simple, however, it may be helpful to point out the three different "systems" on the apparatus figure 2a.

We thank the reviewer for this suggestion. We also inserted brief outlines of the three systems when they are introduced in Section 2.2 (see Lines 165,175,183). The revised Figure 2 and caption are reproduced below.

[Figure]

[1,2,3] **Superscripts indicate configurations for different systems describe in Section 2.2**

**Figure 2: (a) A generalized schematic of the instrument configurations (b) Sample data of simultaneously measured FID and CO$_2$ signals collected on System 1 (i.e., including optional GC)**

Known concentrations of single compounds (with variable O/C) were injected through the apparatus in order to establish that: 1) FID/CO2 correlates with O/C; 2) FID/CO2 correlates with relative effective carbon number; and thus 3), O/C and relative effective carbon number are also correlated. Simple mixtures of 2 organic compounds were analyzed using a GC coupled to the FID+CO2 detector apparatus. The expected FID/CO2 trends nicely with measured FID/CO2. The authors argue that this trend indicates this apparatus can be used in the analysis of atmospheric particles.

The overall presentation is clear and concise. Appropriate high-quality references are made. Results suggest that this apparatus could be successful in the field for O/C, however, additional, more complex testing would make a more convincing case (See Specific Comments #3 and 4). Given the scope of this paper, I do not believe that additional testing is necessary prior to publications. I would recommend further testing prior to publication of field data (either in the lab using more atmospherically relevant conditions/mixtures or comparison tests along side a well established instrument that measures O/C).

Again, we thank for the reviewer for recognizing the value of our approach, as well as the scope of this paper. We are indeed in the process of developing a field-deployable instrument for the measurement of aerosol O/C based on this approach, which will require addressing many of the issues raised.

Specific Comments:

1) In the "Materials" section, complex commercially available mixtures on unknown compounds (perfumes, colognes, etc.) are mentioned. I do not recall any discussion regarding the analysis of these complex unknown mixtures. Did I miss something? If I did not miss something, I am unclear why they are being discussed in the materials section.

We can make this more explicit. We were not interested in commercial mixtures themselves; they only served to provide high numbers of analytes for the later analyses. We have clarified the manuscript as follows (See Section 2.3, lines 207-211).

"A variety of mixtures and individual compounds were analysed in this work. *To facilitate collecting data on a broad range of atmospherically-relevant compounds and compound classes, commercially available mixtures containing* unknown analytes *were analysed*, including four scents of air freshener and six perfumes and colognes. *Analytes in these mixtures were identified by GC-MS. Compounds in these mixtures were analysed only for their FID/CO$_2$ ratios, not to obtain quantitative information on their concentrations. Analytes from these mixtures account for 46 of the total 89 analytes for which FID/CO$_2$ was measured.*"

2) As I read this paper, my initial thoughts went immediately to, "This looks great in the lab, but are the concentrations you used in the lab relevant to the field? Can this apparatus handle the

concentrations you would expect to be measuring in the field?" The SI covers this nicely, but it may be an important to note or reference in the main manuscript as well.

We have added the following clarification to the text of the Introduction (Lines 89-97) and direct the reader to the Supplementary section:

"The high accuracy afforded by a two-cell approach (levels of detection of less than 100 ppb, (LI-COR, 2007)) provides potential detection of FID-produced $CO_2$ at concentrations relevant to *atmospheric applications. One significant potential application of using combined FID and $CO_2$ measurements to measure oxygen content would be the measurement of bulk chemical properties of organic aerosol. As a preliminary assessment of the feasibility of such an approach, we provide in the Supplementary Information (Section S1 and Table S1) an estimate of the concentrations of $CO_2$ generated in the combustion of atmospherically-relevant concentrations of organic aerosols. Using reasonable assumptions for a theoretical instrument based on such an approach, expected* concentrations of $CO_2$ *measured* in the outflow of an FID of ~100 ppb to ~100 ppm, *well within the detection range of current NDIR-based $CO_2$ instrumentation."*

3) It is not clear to me how this apparatus would function in the field. If you are interested in the O/C of atmospheric particulate, how do you plan to deal with the presence of organic gasses entering your apparatus?

We agree that such an apparatus would need to tackle this issue. We mention this in our revised discussion of Application 1 (i.e., field-deployable bulk O/C measurement) (Section 4, lines 452-454):

"*This application would require overcoming additional technical hurdles in sampling and thermal desorption, but approaches have been previously demonstrated for online sampling and thermal desorption of particles (Kreisberg et al., 2009; Williams et al., 2006 Zhao et al., 2013)."*

Members of our research group have demonstrated experience with field-deployable sampling and thermal desorption (please see Kreisberg *et al*. (2009) Aerosol Science and Technology 43: 38-52, doi.org/10.1080/02786820802459583; Zhao *et al.* (2013) Aerosol Science and Technology, 47:258-266, doi.org/10.1080/02786826.2012.747673; Zhao *et al.* (2013) Environmental Science and Technology 47:3781-3787, doi.org/10.1021/es304587x). The specific issue raised by the reviewer of organic gases has been successful treated by this group using an activated charcoal denuder that acts to remove organic gases while permitting particles to pass.

An additional mechanism by which organic gases may impact this approach is that organic gases entering the FID may be combusted and give unwanted signal. However, in contrast to the periodic nature of the thermal desorption of ambient samples, the continuous inflow of low concentrations of background organic gases will result in a fluctuating baseline that will be excluded through peak integration; variability in baselines is a major source for the 10% uncertainty estimate of peak integration.

4) While I recognize the difficulty in replicating atmospheric conditions in the lab, I have a hard time accepting that a mixture of 2 compounds indicates that this method of O/C estimation would be successful in the field. Did you think about testing more complex mixtures? Even a mixture of 2 compounds would provide more adequate evidence that this method would work in the field.

We understand the reviewer's concern about assuming that a two-component mixture provides insight into more complex mixtures. The number of components within a mixture is limited by the two-fold requirement that the analyte mixtures need to have a sufficiently similar vapor pressure to ensure they evolve together during the thermal desorption step and reach the detectors simultaneously, as well as a reasonably large difference in rECNs to test across a wide range. For example, mixtures of octanol (FID/$CO_2$ = 0.92) and dodecane (FID/$CO_2$ = 1.00) or HEM (FID/$CO_2$ = 0.66) and propylene glycol (FID/$CO_2$ = 0.61) would not allow significant dynamic range. Due to the relatively slow temperature ramp rates of the TAG impactor cell and injection port, only compounds with very similar boiling points co-evolve; consequently, the former constraint greatly diminishes the possible constituents of a given mixture. A future cell with a faster possible ramp rate might relax this constraint, but its development requires overcoming a number of technical hurdles otherwise unrelated to this manuscript.

In testing a two-component mixture, our goal was to be sure that a multi-component mixture behaved simply as a mixture of its components. This expectation is supported by the large body of literature on FIDs, which to our knowledge includes no indication that the simultaneous detection of multiple analytes by an FID (e.g., co-elution from a GC) biases the response to one of component. The question reasonably raised by the reviewer is whether a two-component mixture could be extrapolated to a more complex mixture. Two main lines of reasoning support this extrapolation:

1) FIDs exhibit approximately complete combustion and linear response across wide ranges of analyte concentrations, regardless of analyte composition (shown for instance in this work in Figure 3). Consequently, all analytes entering the FID both combust to $CO_2$ and produce linear FID signal, so the resulting signal will represent all analytes with no bias toward some "easier-to-combust" component.

2) The response of a mixture becomes less uncertain as the number of constituents increases. A relevant example of this is the literature on the Aerosol Mass Spectrometer, which uses a fundamental theory of operation that the O/C of all the mass spectral fragments is correlated with the O/C of the parent molecule. In fact, the AMS is calibrated with the fundamentally same approach as this work – as the average relationship of many atmospherically-relevant components. Canagaratna *et al.* (2015, Atm. Chem. Phys.,15, 253-272, doi:10.5194/acp-15-253-2015) found that while the average relationship produced relatively large error for a single component, error decreased with increasing complexity of the mixture, shown in Figure 6 of that reference reproduced below. In other words, as the number of components in a mixture increases, response approaches some average relationship, yielding decreased uncertainties. This suggests that in our system, if the number of components in a mixture were increased, we actually converge more closely toward the general assumption about the correlation between O/C and FID/$CO_2$; a one- or two-component system is the highest uncertainty system. The average relationship to which the system converges may depend in part on the constituents of

the mixtures, so a multi-component mixture created in the lab may still be biased relative to ambient particles

Together, these lines of reasoning suggest that (1) there is no reason the addition of more components would be expected to introduce bias, and (2) increasing complexity generally yields a convergence toward fundamental assumptions and average relationships. Based on these conclusions, the good agreement of a two-component mixture, and the practical hurdles to testing higher-component mixtures, we believe that extrapolation to higher-component mixtures is reasonable. Future comparison to particulate matter will do much more to tease out any biases than creating increasingly complex laboratory mixtures.

[Figure]

" Figure 6a and b show the error in Improved-Ambient O:C and H:C values as a function of the number of standard molecules in the mixture of interest. It is clear from the figure that the error becomes smaller and plateaus for both of the elemental ratios as the number of OA species in the mixture is increased" (Canagaratna *et al.*, 2015, p. 263)

**Figure 6. (a)** Errors in Improved-Ambient O : C ratio of organic standard molecule mixtures as a function of number of species in the mixture. **(b)** Errors in Improved-Ambient H : C ratio of the organic standard molecule mixtures as a function of number of species in the mixture.

(Plot from Canagaratna et al., 2015)

We have included a version of this discussion in the text will be altered as follows (Section 3.2, lines 324-333)

*"Experimental mixtures in this work were limited to two components due to the need for comparable vapour pressures but differing instrument responses. However, extrapolation to more complex mixtures is supported by both the theoretical principles of the approach, and previous work on measurement of O/C by the Aerosol Mass Spectrometer (AMS). That instrument is similarly calibrated as simply the average relationship between analyte O/C and the measured parameter (for the AMS, O/C of molecular fragments) for a large number of individual analytes (Aiken et al., 2008; Canagaratna et al., 2015). Canagaratna et al. found that uncertainty is actually highest in applying the average relationship to one or two components, and decreases with mixture complexity as the average relationship better describes the complex mixture (Canagaratna et al., 2015). The relationship observed between FID/$CO_2$ and analyte O/C can therefore be expected to extend to more complex mixtures, though application of this relationship for bulk measurements of real-world ambient aerosols would need to first be validated."*

5) Methanol was pointed out as an outlier. Are you confident that there are not many more atmospherically relavent "outliers"? Could this potentially skew the trends you are seeing and lead to poor estimations of O/C?

We agree with the reviewer that methanol is a bit concerning. As we cannot reasonably test every atmospheric constituent, we of course cannot discount the possibility there are other outliers. That said, there are mechanistic reasons methanol might be expected to be anomalous for the reasons described by Holm and discussed in the manuscript, and none of the other compounds tested yielded similar anomalous results. Moreover, methanol has the highest opportunity for outlying behavior since it has only a single carbon atom and single functional group; for analytes with more carbon atoms, anomalous behavior of one atom or functional group would have a lower relative effect on FID behavior (i.e., the other atoms would likely still behave normally). An expanded discussion of potential outliers has been included in the revised manuscript (Section 3.2, lines 307-311):

"We speculate therefore that methanol falls off the line as it undergoes a fundamentally different combustion process than all other alcohols, so its behaviour as an outlier does not have negative implications for the application of this system to larger compounds. *Furthermore, any analytes larger than $C_1$ will be less influenced by the anomalous behaviour of one functional group, so any similar outlying behaviour would be less substantial (i.e., other carbon in the molecule would counterbalance the effect)."*

If some other individual analyte does turn out to be an outlier, this would indeed lead to very high error for quantification of that one analyte. However, its impact on uncertainty or error of a bulk measurement of a complex mixture would be proportional to the contribution of that analyte to the mixture. That is, applying the average relationships determined in this work to bulk mixtures would only lead to large error if a large fraction of a mixture were comprised of an outlier. Consequently, as the complexity of the analyzed mixtures increase, the impact of outliers will diminish. A discussion of this effect has been included in new Section 3.4 (lines 391-398):

*"Consequently, the main source of potential bias for atmospheric applications is not error in any specific subset of components, but in whether or not the overall derived relationship accurately describes the mixture to which it is being applied. Confirmation of the appropriateness of the derived equations to any given atmospheric mixture, such as bulk ambient organic aerosol, would require comparing any instrument relying on these relationships to established instrumentation, such as an AMS. We note that the O/C calibration for the AMS is largely based on the same approach used here of developing an average relationship from data for many atmospherically relevant individual analytes and that error of this approach tends to decrease with increasing complexity (Canagaratna et al., 2015), so there is precedent for the successful adoption of this approach."*

6) Is there a reason no nitrogen- or sulfur-containing organics were tested? Would you expect any changes in the FID/CO2 trends in areas rich with these compounds?

The primary concern of this paper is proof-of-concept that O/C relationships correlate with detector responses and the rECN concept. A nitrogen-containing compound, musk ketone, appeared in one of the perfumes; while it approximately obeyed the average relationships, it was excluded as it was the only nitrogen-containing compound in the whole dataset. This is discussed in the revised manuscript (lines 420-422).

Nitrogen and sulfur were not deeply explored here in part because their FID responses are expected to depend on their functional groups (i.e. oxidized versus reduced) and are poorly understood. Literature suggests that a nitrogen bonded to a carbon responds roughly like the analogous alcohol, but little literature is available on the FID response of nitro and nitrate groups. Some applications of the relationships determined in this work are not necessarily impacted by uncertainty around nitrogen- or sulfur-containing compounds, which is one of the reasons we choose not to undergo deep investigation of the various response of different heteroatom functional groups (for instance, using this approach to calibrate a mass spectrometer for low-NOx chemistry). However, other applications obviously do suffer some uncertainty, for instance trying to collect bulk measurements of particle O/C. A detailed discussion of the potential impacts of nitrogen and sulfur has been included in the revised manuscript in Section 3.4, and is discussed here.

Basic assumptions can provide some bounds on the potential uncertainty introduced. It has been shown that C-N bonds behave as an analogous oxygen, so amines or other reduced nitrogen "looks" like an equivalent amount of oxygen in these measurements. On the other extreme is organonitrates, which contain the -$RONO_2$ group; if we assume that the C-O bond suppresses FID response from the carbon atom like an alcohol would (because it is a single C-O bond), and the rest of the atoms do not produce response, this whole group "looks" like a single oxygen atom, despite containing three oxygen atoms. In other words, each nitrogen atom as a nitrate group may serve to "hide" two oxygen atoms. The range of potential impact of nitrogen is therefore between -2 and +1 oxygen per nitrogen; likely the average impact is somewhere in between. In samples with high organic nitrogen or sulfur content, this could lead to significant uncertainty. However, for typical

ambient organic aerosols, in which N/C is usually at least 10 times lower than O/C, this is likely only a 10-20% uncertainty, within the overall uncertainty of the method. We note that this is in fact the same issue the AMS has historically had – that much of the oxygen of organonitrates does not get registered in the O/C calculation and may cause a ~20% uncertainty (Farmer et al., 2010, PNAS 107:6670-6675, doi.org/10.1073/pnas/0912340107). The back-of-the-envelope for sulfur suggests similar effects. To carefully account for the impacts of sulfur or nitrogen in a true field-deployable instrument, one could include additional detectors to measure the scale of the interference (for example, a flame photometric detector tuned for organic sulfur detection).

We reproduce below the discussion included Section 3.4 ("Extension to atmospheric particles"), as well as revisions to the Introduction discussing this topic:

Introduction (lines 98-109)

*"Besides oxygen, nitrogen and sulphur are common heteroatoms occurring in organic aerosols. Both elements are present at concentrations approximately an order of magnitude lower than oxygen or carbon (Aiken et al., 2008; Carrasquillo et al., 2014; Docherty et al., 2011) (Docherty et al., 2011; Surratt et al., 2008), with organic sulphate primarily present as organosulphates, and nitrogen present as a mixture of functional groups including nitrates (Farmer et al., 2010), N-containing heterocyles (Laskin et al., 2015), and amines (Murphy et al., 2007). These compounds also likely efficiently combust in an FID, but the impacts of these functional groups on FID sensitivity of an analyte are poorly studied. The presence of these groups may complicate the relationship between FID response and oxygen content, just as it increases uncertainty in existing measurements of the oxygen content of bulk aerosol (Farmer et al., 2010). Fortunately, the low concentrations of heteroatoms relative to oxygen and carbon in atmospheric aerosols and aerosol components suggests that their overall impacts on most measurement approaches are relatively minor in most atmospheric applications. Consequently, the focus of the present work is on understanding and parameterizing the fundamental relationships between FID signal, $CO_2$ produced in an FID flame, and oxygen content of the analyte(s), and the impacts of other heteroatoms are left for discussion as uncertainties."*

Section 3.4 (lines 401-428)

*"The second issue acknowledges the potential uncertainties caused by the presence of  nitrogen- and sulphur-containing compounds in atmospheric samples. In some applications of these relationships, such issues could potentially be avoided, for instance by limiting analyses to low-$NO_x$ environments, or screening out components that have nitrogen in their molecular formula (e.g., if using this approach to calibrate a mass spectrometer, Application #2 below). However, clearly some applications of these relationships (e.g., bulk analysis or organic aerosol, Application #1 below) require a thoughtful treatment of heteroatoms, so we discuss the issue in some detail here. The effects of nitrogen and sulphur on FID response are less certain than oxygen-containing functionalities. Reduced nitrogen (specifically amines) has been shown to impact FID response similar to alcohols (i.e., single-bonded oxygen), so any amines (and likely other C-N bonds) present in a sample would produce an effect that appeared to be caused by oxygen. Conversely, though nitrate (-$RONO_2$) groups do not have well-studied FID responses, it is reasonable to expect that the C-O bond may have the impact of the C-O bond of an alcohol, and the other heteroatoms (one nitrogen and two oxygens) would likely have no effect on the signal as they are*

*not bonded to a carbon atom. Such a scenario would imply that nitrate groups would bear three oxygens, but cause the effect of only one oxygen, in essence masking two oxygens. These expected impacts suggest that each nitrogen produces signal of between +1 oxygen and -2 oxygen. Sulphur would likely exhibit similar effects. As with other sources of error, these effects may yield high uncertainty for individual components, but for bulk mixtures will be proportional to the amount of nitrogen present, and will likely be mitigated by the presence of a mixture of reduced and oxidized heteroatom-containing groups. Given nitrogen and sulphur are present in atmospheric mixtures at concentrations around ten times lower than oxygen, this would yield only a 10-20% error, which is not substantially beyond the overall uncertainty in the relationships. The specific effect of nitrates "masking" two oxygens was actually previously shown to impact the AMS as well and produce 10-20% error(Farmer et al., 2010), consistent with the analysis here. One nitrogen-containing compound (musk ketone) was studied in this work and did not exhibit significant bias but was excluded from analysis as the relevant structure-activity relationships do not include estimated impacts of the substituent nitro groups. Overall, the complication of heteroatoms in atmospheric applications of these relationships is highly dependent on the application. If used to calibrate individual components that are likely to have high nitrogen content, nitrogen will introduce high uncertainty. However, if applied to bulk ambient aerosols, the error introduced by heteroatoms is on the same scale as existing uncertainty in these relationships. Any user of these relationships should then consider the potential impact of heteroatoms in their specific application. In applications for which heteroatoms need to be accounted for explicitly, the ability to run this detector chain in parallel to other common GC detectors (e.g., a flame photometric detector for sulphur) may allow improved understanding of heteroatom effects."*

Technical Corrections:

1) There is no mention of where the data in Figure 4 (section 3.2) came from. I assume it is from the 90 different compounds that are mentioned in section 2.4.

   Correct. We have clarified the manuscript, Section 3.2, lines 284 as follows:

"Figure 4 shows correlations between three parameters *for the 89 analytes*: FID/CO2 signals, estimated relative ECN, and O/C."

2) Line 35: appears to be space between parentheses and period.

   Corrected.

3) Line 71: two commas after "Generally"

   Corrected.

**Anonymous Referee #2**

This work describes the design and lab testing of a lower-cost and simplified method to determine the carbon and oxygen content of particulate matter. In this manuscript, the authors couple a NDIR CO2 detector to a flame ionization detector to demonstrate a new approach to measure the carbon and oxygen content of atmospherically-relevant organic compounds. Three criteria need are tested through theory and fundamental validation of the approach: 1) FID combustion efficiency 2) FID response per carbon atom (i.e., measured ratio of signals, FID/CO2 ) must be inversely proportional to the oxygen content of analytes and mixtures, 3) achieve high correlation amongst FID/CO2 signal ratios, analyte O/C, and analyte FID sensitivity (rECN) within reasonable error. Field testing is not a component of this first method paper. With some minor improvements, I consider this work worthy of publication.

We thank the reviewer for their time, and appreciate their support in the current scope of the manuscript. We note that we have made explicit this scope in the revised introduction.

Here are some specific comments and questions.

1. Line 71: eliminate one of the commas after "Generally"
   Corrected.
2. Supplement: would be helpful to also list the ECN structure-activity relationship values from the Scanlon & Willis, and Jorgensen publications.
   We have revised the Supplementary Information to include the relevant information from Scanlon and Willis, 1985 and Jorgensen, 1990. These structure-activity relationships and an example of their applications is now included in Tables S2 and S3 and Section S2.

   The main text has also been revised to read (Section 2.1, lines 134-145):

 "In Figure 1, the theoretical slopes of compounds comprised solely of different *substituent* functional groups are shown (dashed lines) as a function of O/C, estimated from existing structure-activity relationships (Scanlon and Willis, 1985). Compounds entirely comprised of carbonyls and carboxyls provide bounding cases, as the carbon atom in both groups produces no FID signal and they add one and two oxygen atoms respectively; the slopes of these relationships are consequently rECN = -O/C and rECN = -0.5*O/C, respectively. Compounds comprised only of alcohols fall in between, with a slope dependent somewhat on the specific structures of the alcohols due to differences in the effects of primary, secondary, and tertiary alcohols (compounds used to estimate slopes shown in Figure 1 are provided in *the Supplementary Information: Section S3, Table S4, and Figure S1).* ECNs for compounds spanning a range of functionalities are available in published literature and are included in Figure 1. *The structure-activity relationships used to calculate rECN, and example calculations are given in Supplementary Information Section S2.* Nearly all compounds fall within the *theoretical* bounding cases as expected, with an average slope approximately in the middle. These data suggest that a direct measurement of rECN *of a compound should generally correlate with its O/C."*

3. Figure 1: The caption should be changed to reflect that dashed line for alcohols represents the rECN calculated for a subset of alcohols, and is not structure independent like with carbonyls and carboxyls.

The caption for Figure 1 has been clarified as shown below:

Figure 1: Plot of relative ECN versus O/C ratios based on literature values. Literature data are from Scanlon and Willis (1985), representing a variety of functional groups shown as different shapes. Dashed lines are the theoretical slopes of compounds comprised completely of labelled functional groups, based on the structure-activity relationship provided by Scanlon and Willis (1985). *In contrast to that of the carbonyls and carboxyls, the theoretical slope of alcohols is not structure independent and is based on a chosen subset of alcohols*. Compounds used to develop the alcohol slope are provided in Table S4.

4. Paragraph starting at line 135: when utilizing a GC/MS system to assist in identification and relating to FID through retention time indices, were there differences in flow rate settings (constant pressure vs constant flow) utilized between the measurement methods, and if so, how was retention time matched/correlated?

We agree this was unclear on the manuscript. While both systems used the same column dimensions, stationary phase, carrier gas, and flow settings, we erroneously stated that both instruments had the same temperature program when in fact there were differences that gave different absolute retention times. To match analytes between systems for identification of FID analytes by MS, relative retention times were converted to an alkane-based retention index using an n-alkane standard (C7-C40). Analytes on the GC-FID generally had nearly identical retention indices on the GC-MS, providing confident matching of analytes. The text is revised in Section 2.2, lines 169-174, as follows:

 *"For identification of unknown analytes in mixtures,* parallel injections were performed on a GC coupled to a quadrupole mass spectrometer (GC-MS) as a detector (7820A/5977, Agilent Technologies) *using the same GC stationary phase, carrier gas, and flow conditions; differences in retention times due to differences in temperature ramps were corrected for by daily injections on each system, of a mixture of n-alkanes ($C_7$-$C_{40}$, Supelco).* Positive identification *of analytes in a mixture of unknowns was* defined as a match in the NIST mass spectral library with strength of at least 850, as well as a retention time in *within the range of retention indices published by NIST (Wallace, 2019)."*

5. Lines 141-142: Please clarify how closely the retention index of a given compound needed to match the NIST values.

NIST values for retention index of a given analyte on a column of a given phase frequently vary by up to 30 units. Positive identification required a retention index within the range provided in the NIST database and consistent with the relative retention times of nearby analytes, as well as a mass spectral match of greater than 850 with the NIST mass spectral library.

The text (Section 2.2, 169-174) has been clarified as described in the previous comment (#4).

6. Paragraph starting at line 135: It would help the reader if a more detailed instrument diagram for systems 1,2(a/b),3 were provided in supplemental information.
   This is addressed in the response to Referee #1 above. Figure 2 has been revised and Section 2.2 (lines 165,175,183) now explicitly describes the instrumental configuration as given below:

"Three instrument configurations were used to analyse a wide range of compounds:

"System 1 *(Agilent Injection Inlet - GC separation - Agilent FID - LI-COR 6262):* Aliquots of 0.2-1.0 µL of mixtures *(including authentic standards, and commercially available fragranced consumer products)* were injected into a heated inlet…""

7. What is the complete list of supporting compressed gases required for this instrument? Helium, Hydrogen, (Nitrogen for FID?), compressed air for FID?

The gases and consumables for the instrument as operated here are described below:

Zero air for FID and LI-COR reference. Hydrogen for FID. Helium as carrier gas and FID make-up gas, if necessary. Zero air generator (ZAG) for desiccating counterflow. For System 2a, Liquid nitrogen was used for cryotrapping.

We note that many of these gases can be generated on-site or otherwise avoided. The absolute minimum list of consumable gases required is: hydrogen for the FID, and inert gas (helium or nitrogen) as carrier gas. With some technical adjustments, we have successfully used compressed room air for the FID, LI-COR reference, and desiccating counterflow.

The text clarification (Section 2.2, lines 148-160) is shown below:

"A schematic of the instrument used to quantify the relationship between FID signal and $CO_2$ produced is shown in Figure 2a. Individual analytes or mixtures were introduced to a thermally controlled cell from which they were thermally desorbed into a helium carrier flow. Analytes were then transferred through heated transfer lines to the FID, using compressed $CO_2$-free zero air as an oxygen source. For some configurations, an in-line GC provided separation of analytes as discussed below. A custom adapter was built to allow tubing to connect the FID outlet to the $CO_2$ detector. To prevent the water produced by the FID hydrogen flame from impairing the carbon dioxide detector, the outflow of the FID was dried using an in-line Nafion permeation dryer (MD-110-48F, Perma Pure LLC) with dry sheath air provided by a zero air generator used as dessicating counterflow. The dried sample stream was detected by a NDIR $CO_2$ analyser (LI-6262 or LI-7000, LI-COR Biosciences), with $CO_2$ -free air provided to the reference cell. For all trials, helium served as the carrier and FID make-up gas where applicable. *In summary, the required supporting gases as operated here include $CO_2$-free zero air (as FID oxygen source and LI-COR reference), zero air for desiccating counterflow, hydrogen (as FID fuel source), and helium (as carrier gas and FID make-up gas, when necessary). Most, or all, of the required air could be supplied as compressed room air with some minor technical adaptations, reducing consumable gas needs to hydrogen and carrier gas.*"

8. Lines 179-183: it's stated the FID/CO2 ratios measured with systems 1&2 were normalized by the ratio of the nearest-eluting alkane. However, with system 2 there was no temperature ramp. Please clarify how the normalization for samples measured by this setup occurred.

As the reviewer notes, Systems 2 and 3 involve injection of single analytes. We have clarified the normalization process in the revised manuscript. Analytes were normalized to an alkane either immediately following or prior to analysis of the analyte of interest; most often, this alkane was dodecane, but choice of reference is not critical because the FID/$CO_2$ ratio was nearly identical for all hydrocarbons.

For System 2A, cryotrapping was needed. Dodecane was dissolved in DCM and injected into the inlet. The cryo system trapped dodecane, whereas the more volatile DCM continued to the detectors. After the DCM evolved, the system was heated, vaporizing dodecane. Octane and decane were also tested with cryotrapping.

For System 2b, injection of a dodecane solution into the room-temperature cell allowed evolution of DCM at room temperature whereas the dodecane only evolved upon heating.

The manuscript is modified as described below (Section 2.4, lines 218-231)

"*Calculation of FID/CO$_2$.* Systems 1 and 2 provided data for *89* analytes, from which the ratio of FID signal/CO$_2$ signal (FID response per CO$_2$ produced) versus O/C ratio could be plotted. Individual compounds were injected at concentrations of approximately 250 ng/µL, either as aqueous solutions or in a carbonaceous solvent purged prior to thermal desorption as described. FID/CO$_2$ of oxygenated and unsaturated compounds were normalized to the FID/CO$_2$ ratio of an *n*-alkane a*nalysed at nearly the same time*. This approach *simplifies interpretation by providing* a value of 1 unit of FID response per CO$_2$ produced for *n*-alkanes (which have the maximum possible FID response), and a value less than or equal to 1 for any oxygenated compound. *This normalization further corrects for any day to day variability in the sensitivity of the FID or drift in the CO$_2$ instrument. In System 1, which involved GC separation, FID/CO$_2$ of an analyte was normalized to that of the n-alkane with the nearest retention time. In System 2 and 3, individually injected analytes were normalized to injections of dodecane immediately preceding or following analysis. The selection of n-alkane for normalization is not critical, as nearly all saturated hydrocarbons were observed to have an FID/CO$_2$ ratio within 1% of the n-alkane average. The FID/CO$_2$ ratio is theoretically independent of the mass of analyte introduced since both FID and CO$_2$ scale with analyte quantity. However, to minimize uncertainty, n-alkanes for normalization were introduced at concentrations similar to the analyte of interest, which should account for any potential non-linearity in detector response.*"

9. Figure 4: it would be helpful to have a table of compounds used in these figures within the supplement. What is the volatility range tested, range of functional groups, or some multifunctional compounds too?

We thank the reviewer for this suggestion. A table (Table S5) and the text below have been added to the Supplementary Information (Section S4):

"*Table S5 gives the functional groups and vapor pressures (via EPI Suite database, US EPA 2019).*

*Of the 89 total compounds, 20 were n-alkanes. Of the remaining 69 compounds, 46 were multifunctional (marked with an asterisk\*). The overall counts of compounds containing a functionality were: 20 alkanes, 18 alkenes, 29 aromatics, 13 ethers, 29 esters, 6 aldehydes, 8 ketones, 35 alcohols, 1 acid, 1 epoxide, 2 lactones (also counted as esters) and 2 polycyclic aromatics (also counted as aromatics). Therefore, pentaerythritol counts as 1 alcohol, whereas hydroxyacetone is counted as both a ketone and an alcohol.*

*The volatility range (through EPI) was 3.38 E-6 to 8120 Pa (pentaerythritol to ethanol).*"

10. Lines 236-237: The sentence "It is notable but probably incidental that the observed downward slopes and R2 values match closely with those of a set of alcohols, -0.58." seems ambiguously worded, and relies too heavily on the reader to have read section S2 in the supplement to understand what is meant.

We agree that the original statement is unclear and does not contribute substantially to the paper. This comment has been removed from the revised manuscript.

11. Paragraph starting at line 243 and Figure 5: why did the authors select these specific mixtures?

The mixtures were chosen based on three criteria:

1) Both components had already been run individually so that their $FID/CO_2$ was well-characterized.

2) The compounds' vapor pressures were similar enough that they would evolve from the cell simultaneously, and thus be detected as a mixture. This constraint is fairly stringent, since the relatively slow temperature ramp of the TAG cell means that only compounds with very similar boiling points co-evolve.

3) There needed to be a large enough difference in the two components' $FID/CO_2$ so that mixtures of varying composition would have large differences to be measurable with some certainty. For instance, a mixture of two co-evolving compounds with only moderate differences in $FID/CO_2$ (e.g., dodecane and octanol) would not provide significant dynamic range.

(For additional detail, see response to Referee #1, Comment #4 above)

The revised manuscript includes a more detailed discussion of these criteria and the justification for extrapolation between two components (Section 2.4, lines 324-333):

"*Experimental mixtures in this work were limited to two components due to the need for comparable vapour pressures but differing instrument responses. However, extrapolation to more complex mixtures is supported by both the theoretical principles of the approach, and previous work on measurement of O/C by the Aerosol Mass Spectrometer (AMS). That instrument is similarly calibrated as simply the average relationship between analyte O/C and the measured parameter (for the AMS, O/C of molecular fragments) for a large number of individual analytes (Aiken et al., 2008; Canagaratna et al., 2015). Canagaratna et al. found that uncertainty is actually highest in applying the average relationship to one or two components, and decreases with mixture complexity as the average relationship better describes the complex mixture (Canagaratna et al., 2015). The relationship observed between $FID/CO_2$ and analyte O/C can therefore be expected to extend to more complex mixtures, though application of this relationship for bulk measurements of real-world ambient aerosols would need to first be validated.*"

12. Line 269: why would we want to determine rECN when we already have O:C from measured FID/CO2, which would be the info utilized by atmospheric scientists.

The relationships in this manuscript between $FID/CO_2$, rECN, and O/C suggest that with any one of these parameters, the two can be estimated. We agree with the reviewer that perhaps the most obvious

application of the relationships in the manuscript are to use measured $FID/CO_2$ to determine O/C (as mentioned, an application currently under development by this group). However, a number of different atmospheric applications may make use of other pairs of parameters, some of which are described in the "Applications" section of the manuscript.

The case specifically relevant to the question raised by the reviewer is Application 2, the use of an FID to calibrate a novel mass spectrometer. FID response factor per mole of an analyte is described by its ECN (rECN * carbon number). In cases where a formula (and thus number of carbon and oxygen atoms) is known, but not a structure, the relationship in this manuscript between rECN and O/C can be used to estimate FID response and account for the influence of oxygen without the need to measure the $FID/CO_2$ ratio. This is, for example, potentially useful for understanding response of chemical ionization mass spectrometers being widely used by the atmospheric community, but for which calibration remains an open question (e.g., the iodide CIMS). As a case study, imagine analysis of an unknown analyte with formula $C_{12}H_{20}O_2$ (O/C =0.167) that happens to be linalyl acetate. Using the slope equation of Figure 4c (rECN = -0.61(O/C) +0.99), the calculated rECN of this analyte would be 0.89 and thus the ECN = 10.7 (i.e., 0.89 * 12), versus the structure-based estimate of ECN of 10.8 (per Scanlon and Willis: ECN of ester = carbon number – 1 and each olefinic carbon reduces response by 0.05). Without knowing the structure, the FID response factor is therefore estimated within 1%.

One could alternately imagine using an $FID/CO_2$ measurement to estimate rECN (FID response per carbon) for analytes for which no identification, structure, formula, or O/C is known. This group currently has no plans to apply these relationships in this way, but such an application is nevertheless potentially useful for atmospheric mixtures (in which a large fraction of GC analytes frequently have no known identification).

The utility of this potential application has been clarified in Application 2, and alluded to in a few locations throughout the revised manuscript:

(Introduction, lines 71-79)

"*For applications in which* molecular structures of *individual* analytes are known (*e.g., quantification by chromatographic instruments)*, the FID sensitivity of an analyte can be calculated from established structure-activity relationships (Scanlon and Willis, 1985). Generally, a carbon with a carbonyl does not produce any FID signal, and a carbon with a hydroxyl group produces half as much FID signal as a hydrocarbon. This relationship can be quantified more precisely as the Effective Carbon Number (ECN) of a compound, which describes FID response as equivalent to a hydrocarbon of a certain carbon number. Operating at ambient pressures and requiring only a source of hydrogen, the FID is consequently a stable and low-cost alternative to mass spectrometry, providing robust quantification but with substantially reduced chemical resolution. *However, accounting for the impact of oxygen-containing function groups on FID response currently relies on either knowing molecular structures, or catalytic conversion of all carbon.*"

(Section 3.3, lines 357-360)

"Generally, the rECN can be predicted from either $FID/CO_2$ (Figure 6b) or O/C (Figure 6c) with errors of 10-15% for highly oxygenated compounds *(consistent with propagated error estimates)*, and <5% for less oxygenated compounds (O/C < ~0.5). These low errors indicate that the sensitivity of an FID can be

estimated with high certainty either directly from O/C or, in the absence of this information, from a direct measurement of $FID/CO_2$."

(Section 4, lines 459-468)

"Application 2 An FID could be used as a calibration tool for new instrumentation *without requiring molecular structural information to estimate FID response*. While the FID is an attractive near-universal detector, the structural dependence of its response has limited its adoption. However, we demonstrate here that FID sensitivity can be robustly estimated with low uncertainty from O/C or measured $FID/CO_2$ ratio. This implies that any instrument that can be coupled to an FID can use it for quantification, regardless of whether a molecular formula is available. For instruments that do provide a molecular formula, quantification by FID is possible even without including the additional complexity of a $CO_2$ analyser. *For example, by applying the relationship between O/C and rECN, an FID in parallel with a chemical ionization mass spectrometer could use the molecular formula from the mass spectrometer to estimate FID sensitivity. This could allow improved understanding of response or sensitivity of new atmospheric measurement approaches.*"

13. The technique shows good promise, it will be interesting to see how it performs on ambient mixtures in the field, relative to the existing high-cost, high-maintenance techniques.

We thank the reviewer again for their comments, and similarly hope the relationships in this manuscript prove to be applicable to bulk ambient mixtures. Our next steps are indeed to push these fundamental relationships toward application to atmospheric particles by tackling some of the other technical hurdles described.

**Anonymous Referee #3**

I am puzzled by the disconnect between what the paper says is a main goal – to develop a fast/easy/inexpensive way to measure organic carbon and O/C in particulate matter, and the fact that the authors present no measurements of particulate samples. The paper does describe a method for measuring the carbon and oxygen content of organic compounds that could potentially be used for particulate samples, but how that application would work is not explained and certainly not demonstrated. The method involves a combination of flame ionization (FID) and carbon dioxide (CO2) detection schemes, and the ratio of FID to CO2 response is used to estimate the O/C ratio of organic species. The paper describes a potentially interesting and useful technique, but there are major issues with the paper, and it will need considerable work before it is acceptable for publication.

We thank the reviewer for their comments and their recognition of the fundamental goal of this paper. The scope of the present manuscript is to demonstrate the fundamental relationships between the measurable parameters O/C, FID per-carbon response factors (rECN), and measured FID/CO$_2$ ratios. There are a wide range of ways that these fundamental relationships could be applied to atmospheric systems, and each carries with it a different set of technical hurdles that would need to be overcome and potential uncertainties. As such, it is our belief that the scope of the current manuscript is reasonable, as it is limited only to demonstrating the underlying relationships and includes a discussion of some of the ways that they could applied. We note that the abstract focuses primarily on the underlying relationships and the conclusions thereof, and only notes the application to particle carbon as a possibility. The obvious application of this approach is the estimation of bulk organic aerosol O/C by measuring FID/CO$_2$, and as such that application is discussed somewhat throughout the manuscript (and is currently in development by this research group). However, developing such an instrument, sampling and analyzing real-world particulate matter, and accounting for nitrogen and sulfur, are issues specific to that application and not necessarily germane to the fundamental relationships in this manuscript. We consequently feel that tackling and/or overcoming those issues are beyond the scope of the present work; however, we agree that many readers will have similar such questions. We consequently have revised the manuscript to make the scope of the present work more clear and include a discussion of some of the hurdles and uncertainties that may (depending on application) need to be addressed to apply this approach to atmospheric systems.

In particular, we highlight the scope of this paper, and the implications of heteroatoms, in the Introduction (lines 98-112):

"*Besides oxygen, nitrogen and sulphur are common heteroatoms occurring in organic aerosols. Both elements are present at concentrations approximately an order of magnitude lower than oxygen or carbon (Aiken et al., 2008; Carrasquillo et al., 2014; Docherty et al., 2011) (Docherty et al., 2011; Surratt et al., 2008), with organic sulphate primarily present as organosulphates, and nitrogen present as a mixture of functional groups including nitrates (Farmer et al., 2010), N-containing heterocyles (Laskin et al., 2015), and amines (Murphy et al., 2007). These compounds also likely efficiently combust in an FID, but the impacts of these functional groups on FID sensitivity of an analyte are poorly studied. The presence of these groups may complicate the relationship between FID response and oxygen content, just as it increases uncertainty in existing measurements of the oxygen content of bulk aerosol (Farmer et al., 2010). Fortunately, the low concentrations of heteroatoms relative to oxygen and carbon in atmospheric aerosols and aerosol components suggests that their overall impacts on most measurement*

*approaches are relatively minor in most atmospheric applications. Consequently, the focus of the present work is on understanding and parameterizing the fundamental relationships between FID signal, $CO_2$ produced in an FID flame, and oxygen content of the analyte(s), and the impacts of other heteroatoms are left for discussion as uncertainties. There are a number of potential atmospherically-relevant applications of these relationships, including calibration of instruments for which molecular structures of analytes are not known, and bulk analysis of oxygen content of ambient or laboratory-generated organic aerosol."*

I have the following general and specific comments that need to be dealt with.

General Comments: No particle-phase, or particulate-like samples were analyzed in this work.

This paper is that demonstrates the correlations between O/C ratios, $FID/CO_2$ ratios, and the Relative Effective Carbon Number (rECN), for potential use in a number of atmospherically-relevant applications. The application the reviewer suggests is obviously of interest and will require overcoming technical hurdles around sampling and sample transfer. However, we note that other atmospherically-relevant applications of the relationships in the manuscript may not need to overcome these hurdles. For example, in Application 2 in the manuscript we propose using molecular formulas of analytes to estimate FID response of individual analytes without the need to know their structure, which could be useful for understanding the response of chemical ionization mass spectrometers being deployed for atmospheric measurements (e.g., by coupling with a GC, or introducing individual analytes). Such an application would not necessarily require analysis of particle-like samples. This highlights our belief that while sampling particulate matter is an obvious application of the relationships in this work, it is not the only one. We agree that such an instrument would need to be demonstrated for particulate matter through comparison to other instrumentation, but we believe this is beyond the scope of describing the underlying relationships (and quantifying the uncertainty of their application).

If the authors were to try to emulate particle organic carbon (POC) they would realize they should also be working with compounds like oxalic acid and glyoxal, which have O/C ratios of 2 and 1, but probably no FID response. There is potentially a lot of oxalic acid/oxalate – which would represent a large deviation from the clusters of points that are presented by the compounds studied in this work. There needs to be some discussion of how such compounds would affect what is actually observed for POC.

We agree that application of these relationships to atmospheric conditions or organic aerosol warrant additional discussion, which we have included in the Results and Discussion section, excerpted at the end of this comment. We discuss here in detail the issues of potential outliers with large expected atmospheric contributions, like oxalic acid and glyoxal.

We also agree with the reviewers that compounds such as glyoxal and oxalic acid would give negligible signal on FID (i.e., an rECN near zero). We note that oxalic acid, with O/C = 2 and expected $FID/CO_2 = 0$ (the more atmospherically dominant of the two mentioned analytes) actually does not fall very off the average line. The application of the relationships in this manuscript for measurement of these

compounds would introduce some error, but the error depends strongly on how the relationships are being applied. If $FID/CO_2$ were being measured for oxalic acid as an individual analyte (e.g., eluting from some chromatographic separator), the amount of $CO_2$ measured would provide the amount of oxalic acid, and the lack of any FID signal would indicate an O/C of approximately 1.84 (<10% error). However, glyoxal would behave approximately similarly, which would give much higher error and would make the two compounds mostly indistinguishable.

However, when considering the application of these relationships to the measurement of bulk aerosol, it is critical to note that the error introduced from any one compound is proportional to contribution that compound to the bulk. This is one of the reasons that the uncertainty in the measurement decreases as more compounds are added and complexity increases (see Referee #1, comment #4 above); essentially, while the best fit line describes the average relationship well, it describes any individual analyte only with moderate uncertainty. In cases with very large contributions of glyoxal or oxalic acid, error will start to approach their individual errors, but these cases are not expected to be common in real-world atmospheres.

We explore here the theoretical impact of adding a large amount of oxalic acid to our data by assuming that 10% of all the data is comprised of oxalic acid, with presumed values of O/C=2.0, rECN = 0 and $FID/CO_2$ = 0. We show below the influence of oxalic acid on the overall data set, when oxalic acid is weighted 10 times more than the other 89 analytes; this simulates a mixture comprised of 10% oxalic acid and the rest distributed equally across the other analytes. Overall, the impact of this large fraction of oxalic acid is to shift the slope of the $FID/CO_2$ vs O/C relationship by ~7%; this is within our stated uncertainty as described in Section 3.3. In other words, the presence of a large fraction of oxalic acid in a bulk mixture does not significantly bias the average relationships away from that determined in this manuscript. This result is in large part due to the fact that oxalic acid actually reasonably obeys the relationship expected – a large contribution of glyoxal would yield some uncertainty. However, a brief literature search suggests that 10% w/w oxalic acid would be unusually high. Across a variety of urban, rural and remote sites on several continents over a twenty-year period, oxalic acid seems to typically contribute ~5% or less of total organic carbon, with glyoxal around an order of magnitude lower. Consequently, while the uncertainty of these compounds may introduce error in applications where they are quantified as individual analytes or comprise the majority fraction of a mixture, they are expected to introduce at most a few percent of uncertainty or bias in applications to typical ambient aerosols.

[Khwaja et al., 1994 (Atmospheric Environment 29(1): 127-139)), doi.org/10.1016/1352-2310(94)00211-3; Zhou et al., 2015 (Journal of Geophysical Research: Atmospheres 120:9772-9778), doi.org/10.1002/2015JD023531; Rohrl and Lammel, 2001 (Environmental Science and Technology 35(1): 95-101), doi.org/10.1021/es0000448; Sempere and Kawamura, 1994 (Atmospheric Environment 28(3):449-459), doi.org/10.1016/1352-2310(94)90123-6; Limbeck, 1999 (Atmospheric Environment 33:1847-1852),doi.org/10.1016/S1352-2310(98)00347-1; Nah et al., 2018 (Atmospheric Chemistry and Physics 18:11471-11491), doi.org/10.5194/acp-18-11471-2018; Garcia-Alonso et al., 2006 (Toxicological and Environmental Chemistry 88(3):445-452), doi.org/10.1080/02772240600796837].

The table below describes the changes in the slope and intercept of the Figure 4 plots if 10% w/w oxalic acid is added to the existing dataset.

| Fit equations | Data | Data+OxAc(10%) |
|---|---|---|
| 4a. $FID/CO_2$ vs O/C | $FID/CO_2 = -0.54(O/C) + 0.99$ | $FID/CO_2 = -0.50(O/C) + 0.98$ |
| 4b. $FID/CO_2$ vs rECN | $FID/CO_2 = 0.85(rECN) + 0.14$ | $FID/CO_2 = 0.98(rECN) + 0.02$ |
| 4c. rECN vs O/C | $rECN = -0.60(O/C) + 0.99$ | $rECN = -0.50(O/C) + 0.98$ |

The plots are given below. Methanol and glyoxal are also shown but are not included in the fitted slope.

[Figure]

The revised manuscript explicitly addresses the possibility that atmospheric mixtures may not be perfectly described the analyzed set of compounds, and highlights many of the discussion points above (Section 3.4, lines 383-400):

"*The first issue recognizes that the relationships derived are quantitatively described by the weighted average of the set of analytes investigated. If a compound or mixture analysed is not well described by this set of analytes, it may bias application of the average relationship. For instance, some major atmospheric constituents are not expected to obey the average relationships (e.g., glyoxal, O/C = 1.0, rECN = 0), so, as discussed, error is expected in the application of these relationships to individual analytes. However, for individual components, error is on average 10-20%, and in cases where high accuracy is needed for only one or two components, the relationships here are probably not the preferred approach. For complex mixtures, such high-error compounds will only introduce error to the extent that they are also major contributors to the mixture; in most cases, atmospheric samples are sufficiently complex that any error for one component would not introduce significant error for a mixture. Consequently, the main source of potential bias for atmospheric applications is not error in any specific subset of components, but in whether or not the overall derived relationship accurately describes the mixture to which it is being applied. Confirmation of the appropriateness of the derived equations to any given atmospheric mixture, such as bulk ambient organic aerosol, would require comparing any instrument relying on these relationships to established instrumentation, such as an AMS. We note that the O/C calibration for the AMS is largely based on the same approach used here of developing an average relationship from data for many atmospherically relevant individual analytes and that error of this approach tends to decrease with increasing complexity (Canagaratna et al., 2015), so there is precedent for the successful adoption of this approach. Furthermore, analysis of ambient aerosols has indicated that on average atmospheric oxidation adds approximately equal parts double-bonded (e.g., carbonyl) and single-bonded oxygen (e.g., hydroxyl), which would be expected to yield an expected average relationship between FID/$CO_2$ and O/C reasonably similar to that measured here.*"

In addition, organic nitrates and organo-sulfates should also be explored if the authors really want to demonstrate that this technique works for POC. The only hetero-atom (i.e. non-O atom) containing species, Musk ketone (apparently a di-nitro compound) was excluded from the analysis – because it did not conform. This does not bode well for the technique. What was the nature of the problem that led to this disqualification, and how does that bear on whether the technique can work for nitro-aromatics for example?

As the reviewers note, there are complications in applying the demonstrated relationships to ambient atmospheric particles. We address these concerns in a revised discussion section highlighting some technical hurdles. Musk ketone was excluded because it was the only compound containing a heteroatom and structure-activity relationships for oxidized nitrogen-containing functional groups are not well constrained (so it could not be reasonably assigned a structure-based rECN). We note that the measured FID/$CO_2$ actually roughly conforms with its O/C, and it is not a significant outlier. Specifically, the observed FID/$CO_2$ ratio (0.844) would suggest an O/C of 0.270, versus the true value of 0.357 for a 24% error. Most individual analytes fall within 20% uncertainty, so the error is not excessive. The revised

manuscript includes a mention of musk ketone and the reason for its exclusion (Section 3.4, lines 420-422).

The quantitative analysis in the current manuscript is limited to oxygen as a heteroatom for a few reasons. (1) The impact of nitrogen on rECN is likely to depend strongly on the functional group in which the nitrogen is contained, so a detailed exploration of this effect would require a much more in-depth look at a wide range of nitrogen-containing groups, many of which do not have many commercially-available options, and do not have well established impacts on ECN. (2) Some applications of the fundamental relationships in this manuscript may not be strongly impacted by nitrogen or other heteroatoms (e.g., Application 2, calibrating a chemical ionization mass spectrometer, in which case hetero-atom containing species can be excluded on the basis of formulas), so these relationships are useful even without tackling this additional variable. (3) One option for accounting for heteroatoms in the application of an instrument to use FID/CO2 to measure O/C may be the addition of other parallel detectors, which would require overcoming technical hurdles beyond the scope of this manuscript.

For these reasons, we do not attempt to quantitatively account for all heteroatom-containing functional groups and include in the revised manuscript a discussion of this issue. In our response to Referee #1, Comment #6, above, we estimate that the presence of nitrogen in ambient particles would introduce 10-20% error in O/C (in the absence of other detectors or other means to correct for this effect), which is near the stated uncertainty of the approach. We note that the AMS has previously been found to have an approximately similar uncertainty in O/C driven by the presence of organonitrates, which have historically been not well-accounted-for in their approach. We leave the issue of heteroatoms in bulk mixtures for future work that would focus on implementing Application 1, a bulk measurement of organic aerosol. The new discussions of heteroatoms in the Introduction and discussion of Extension to Atmospheric Samples (Section 3.4) are excerpted below.

Introduction (lines 98-112)

"*Besides oxygen, nitrogen and sulphur are common heteroatoms occurring in organic aerosols. Both elements are present at concentrations approximately an order of magnitude lower than oxygen or carbon (Aiken et al., 2008; Carrasquillo et al., 2014; Docherty et al., 2011) (Docherty et al., 2011; Surratt et al., 2008), with organic sulphate primarily present as organosulphates, and nitrogen present as a mixture of functional groups including nitrates (Farmer et al., 2010), N-containing heterocyles (Laskin et al., 2015), and amines (Murphy et al., 2007). These compounds also likely efficiently combust in an FID, but the impacts of these functional groups on FID sensitivity of an analyte are poorly studied. The presence of these groups may complicate the relationship between FID response and oxygen content, just as it increases uncertainty in existing measurements of the oxygen content of bulk aerosol (Farmer et al., 2010). Fortunately, the low concentrations of heteroatoms relative to oxygen and carbon in atmospheric aerosols and aerosol components suggests that their overall impacts on most measurement approaches are relatively minor in most atmospheric applications. Consequently, the focus of the present work is on understanding and parameterizing the fundamental relationships between FID signal, $CO_2$ produced in an FID flame, and oxygen content of the analyte(s), and the impacts of other heteroatoms are left for discussion as uncertainties. There are a number of potential atmospherically-relevant applications of these relationships, including calibration of instruments for which molecular structures of analytes are not known, and bulk analysis of oxygen content of ambient or laboratory-generated organic aerosol.*"

(Section 3.4, lines 401-428 )

*"The second issue acknowledges the potential uncertainties caused by the presence of nitrogen- and sulphur-containing compounds in atmospheric samples. In some applications of these relationships, such issues could potentially be avoided, for instance by limiting analyses to low-$NO_x$ environments, or screening out components that have nitrogen in their molecular formula (e.g., if using this approach to calibrate a mass spectrometer, Application #2 below). However, clearly some applications of these relationships (e.g., bulk analysis or organic aerosol, Application #1 below) require a thoughtful treatment of heteroatoms, so we discuss the issue in some detail here. The effects of nitrogen and sulphur on FID response are less certain than oxygen-containing functionalities. Reduced nitrogen (specifically amines) has been shown to impact FID response similar to alcohols (i.e., single-bonded oxygen), so any amines (and likely other C-N bonds) present in a sample would produce an effect that appeared to be caused by oxygen. Conversely, though nitrate ($-RONO_2$) groups do not have well-studied FID responses, it is reasonable to expect that the C-O bond may have the impact of the C-O bond of an alcohol, and the other heteroatoms (one nitrogen and two oxygens) would likely have no effect on the signal as they are not bonded to a carbon atom. Such a scenario would imply that nitrate groups would bear three oxygens, but cause the effect of only one oxygen, in essence masking two oxygens. These expected impacts suggest that each nitrogen produces signal of between +1 oxygen and -2 oxygen. Sulphur would likely exhibit similar effects. As with other sources of error, these effects may yield high uncertainty for individual components, but for bulk mixtures will be proportional to the amount of nitrogen present, and will likely be mitigated by the presence of a mixture of reduced and oxidized heteroatom-containing groups. Given nitrogen and sulphur are present in atmospheric mixtures at concentrations around ten times lower than oxygen, this would yield only a 10-20% error, which is not substantially beyond the overall uncertainty in the relationships. The specific effect of nitrates "masking" two oxygens was actually previously shown to impact the AMS as well and produce 10-20% error (Farmer et al., 2010), consistent with the analysis here. One nitrogen-containing compound (musk ketone) was studied in this work and did not exhibit significant bias but was excluded from analysis as the relevant structure-activity relationships do not include estimated impacts of the substituent nitro groups. Overall, the complication of heteroatoms in atmospheric applications of these relationships is highly dependent on the application. If used to calibrate individual components that are likely to have high nitrogen content, nitrogen will introduce high uncertainty. However, if applied to bulk ambient aerosols, the error introduced by heteroatoms is on the same scale as existing uncertainty in these relationships. Any user of these relationships should then consider the potential impact of heteroatoms in their specific application. In applications for which heteroatoms need to be accounted for explicitly, the ability to run this detector chain in parallel to other common GC detectors (e.g., a flame photometric detector for sulphur) may allow improved understanding of heteroatom effects."*

The LI-COR instruments used for CO2 measurements have distinct characteristics: the LI-COR 6262 is an older model and has an inherently non-linear response to CO2 that must be calibrated throughout the useful range with multi-point calibrations, and the LI-COR 7000 has a linearized response. The specifics of how these two different instruments were calibrated need to be presented. In addition, these

detectors are concentration-sensitive and therefore the systems, at least as far as I can understand them from the diagram, require an accurate knowledge of the flow rates and pressures in the systems. The details of how these were determined need to be presented in this work.

We have added detailed discussions of flows and calibrations, excerpted below. All flows are regulated by mass flow controllers (for calibration and desorption flows) and electronic pressure controllers (for FID flows), so flows and pressures are well-controlled and stable.

(Section 3.3, lines 336-345)

"The major benefit of quantifying the relationships between O/C, rECN, and measured FID/$CO_2$ is their potential use in predicting one parameter from another. *To understand the error in such a prediction, it is first useful to evaluate how precisely any of these parameters are known. O/C is known precisely for each analyte, and published uncertainties in ECN are on the order of 10% (Faiola et al., 2012). Uncertainty in FID/$CO_2$ is, practically speaking, dominated by uncertainty in the data analysis. All flow rates and pressures are controlled and known to within 2%, and precision of the $CO_2$ detector is similarly negligibly small. Because FID/$CO_2$ is theoretically concentration independent, uncertainties in detector calibration and injection volumes are unimportant, and are low in any case (10% and ~5% respectively). FID and $CO_2$ data can therefore be generated precisely, and uncertainty is primarily driven by analysis, specifically the estimated 10% uncertainty in peak integration (Isaacman-VanWertz et al., 2017). Calculating FID/$CO_2$ requires integrating two different peaks, so combined uncertainty in this parameter is ~15%."*

(Section 2.4, lines 235-240; particular to complete combustion experiments. See discussion below.)

"To assess combustion efficiency, a known mass flow of $CO_2$ gas (2 *± 0.2* sccm of 1% $CO_2$ in balance air) was introduced to the desorption cell in the same location as the injection of analytes to provide a signal-to-mass response factor for the $CO_2$ analyser. As all flows and pressures are controlled electronically, this flow of known calibrant undergoes the same flow and pressure conditions as any desorbed analyte. *Uncertainty in calibrant flow (10%, due to operating the mass flow controller at the low end of its full scale) and $CO_2$ dominates over other sources of uncertainty (e.g. analyte mass injected) in this calibration."*

As discussed in the text, to calculate FID/$CO_2$ ratios, an absolute calibration of the $CO_2$ instrument is not actually required. The FID/$CO_2$ ratios of analytes are normalized to that of alkanes, so is unitless and is impacted by the precision, but not the accuracy, of both the FID and the $CO_2$ measurement. This assumption does require the $CO_2$ from an analyte to have the same response factor as the $CO_2$ from the alkane; given that the species measured ($CO_2$) is the same in both cases assumption is reasonable for linear instruments, and is reasonable for non-linear instruments so long as the analyte and the alkane are introduced at similar concentrations. In the revised manuscript, we have clarified that this was the case.

The only data presented in the manuscript for which absolute calibration of the $CO_2$ instrument is required is Figure 3, demonstrating approximately complete combustion. The revised manuscript includes a detailed discussion of uncertainties, discussed here. For these data, the $CO_2$ instrument was calibrated under the instrument flow conditions. A small flow (2 sccm, controlled by a mass flow

controlled) of 1% $CO_2$ was introduced into the desorption flow (18 sccm, controlled by a mass flow controller) and passing through the FID (230 sccm of air+hydrogen, controlled by an electronic pressure controller), mirroring the flow and pressure conditions of desorbed analytes. This approach provided a constant level of instrument response (LICOR 7000, in the case of these experiments) to a known molar flow rate of $CO_2$. While this known molar flow rate could be assigned a known concentration based on measured flow rates (roughly 80 ppm), it is not strictly necessary to do so, since the molar flow rate of $CO_2$ is known; instead this approach essentially provides a $CO_2$ response factor (instrument response per molecule/s) that was used to calculate $CO_2$ in the desorption flow during analyte desorption.

Inherent uncertainty in O/C is simply an unavoidable feature of the technique, but we don't have any substantive discussion that places this in context. How do the uncertainties in O/C ratios from this technique compare to uncertainties from AMS measurements of O/C? Does this technique really represent an advance in this area or is its' main feature that it is less expensive and easier to field?

(See Referee #1, Comment #4 above)

We thank the reviewer for suggesting a more detailed comparison to the AMS, and a more thorough examination of error. We have revised the manuscript to include this (please see Section 3.2, lines 324-333), and provide additional detail below.

The errors in our approach and the AMS are in fact similar. We note that the AMS approach to measuring O/C was calibrated in much the same way as our approach here – individual analytes were introduced, from which an average relationship was fit to relate mass spectral fragment O/C to analyte O/C. Below, we compare our Figure 6a to data from Canagaratna et al., 2015 (graciously provided by Dr. Canagaratna). Similar levels of error are observed in applying either method to individual analytes, with similar heteroscedastic trends of higher relative error at lower O/C. The errors of our method for lower O/C values are a bit lower, but at higher O/C the errors are comparable and converge. As noted by the reviewer, using this approach to measure O/C would be considerably less expensive and easier to field than an AMS (though of course, it would also provide less chemical detail). A few comparisons between the AMS and the approach in this manuscript have been added throughout the revised manuscript and are excerpted at the end of this comment.

(Section 3.3, lines 348-357)

"Error in predicted values is less than 20% in nearly all cases, *suggesting error in this relationship is not dominated by uncertainty in parameters themselves, but rather inherent error is applying the average relationships to individual analytes*. However, error is often much lower and exhibits some heteroscedasticity worth discussing. Generally, absolute error is higher for oxygenated components due to the structurally-dependent effects of oxygen, which are ignored in these relationships (e.g., the divergence of the carbonyl and carboxyl trends). However, prediction of O/C from FID/$CO_2$ at low O/C yields very high relative error despite low absolute error because as O/C approaches 0, even low absolute errors imply high relative error. For highly oxygenated compounds, relative error in O/C appears to plateau to about 20%; *this level of uncertainty for individual analytes is comparable to that of*

*the AMS. Error in estimating O/C from FID/CO₂ for an individual analyte can therefore* be reasonably summarized as a relative error of 20% with a minimum absolute error of approximately 0.05."

(Section 3.4, lines 418-420)

*"The specific effect of nitrates "masking" two oxygens was actually previously shown to impact the AMS as well and produce 10-20% error (Farmer et al., 2010), consistent with the analysis here."*

**Error in O/C predictions**

From FID/CO₂, Figure 6a of this manuscript

[Figure]

[Figure]

AMS, data from Canagaratna et al., 2015

[Figure]

[Figure]

Comparison (only quantiles shown)

[Figure]

[Figure]

Specific Comments:

Page 4, Lines 119-120. This statement needs much more context given that the slopes of rECN to O/C can vary between -1 and -0.5. Is this really good enough for the purposes of what we'd like to do with POC data?

We are not fully clear on the question being posed by the reviewer. While any individual compound is more-or-less guaranteed to fall along a slope of between -1 and -0.5, a central outcome of this manuscript is that a slope of approximately -0.54 describes the average relationship reasonably well, with a typical error for individual compounds of ~20% for high O/C, and ~0.05 units for lower O/C.

An important consideration in applying this relationship to bulk organic aerosol, is that an average relationship actually describes a complex mixture better than it does an individual analyte. For instance, while the average relationship may underestimate O/C of one analyte, it may overestimate O/C of another, so that statistically the uncertainty actually decreases with increasing number of analytes. This idea was nicely demonstrated by Canagaratna et al. in the same work referenced above, and similarly relying on an average relationship to estimate O/C. Their Figure 6 is reproduced below, and shows that while uncertainty in O/C estimates are highest for individual analytes (on average 28% as stated in that reference), uncertainty in O/C drops by a factor of 2-3 as number of components in a mixture increases and the underlying relationships approach the average. A discussion of tendency for a complex mixture to approach the underlying average relationships has been added to the manuscript (see lines 327-333,365-371).

[Figure]

**Figure 6. (a)** Errors in Improved-Ambient O:C ratio of organic standard molecule mixtures as a function of number of species in the mixture. **(b)** Errors in Improved-Ambient H:C ratio of the organic standard molecule mixtures as a function of number of species in the mixture.

Page 5. The instrument diagrams and associated descriptions are not nearly adequate to explain what was done, where uncertainties could be introduced, and how POC sampling and analysis would work. Doesn't the TAG cell have a filter in it? Was that part of the configuration, if not, how would that change how the system would operate?

We have revised Figure 1 and Section 2 to include more specific information about instrument configures, and a more comprehensive discussion of error is given in the expanded Section 3.3. We note that the instrument diagram in this manuscript is designed to portray the instrument setup for the data presented in this manuscript, not an application of these fundamental relationships to a field-deployable instrument.

In a few locations throughout the revised manuscript, we have added a detailed discussion of uncertainty. These sections are excerpted at the end of this comment and discussed here. In brief, uncertainty is dominated by uncertainty in integration of desorption/chromatographic peaks. In the case of the complete combustion experiments, a similar level of uncertainty is introduced in the flow rate of the $CO_2$ gas used to calibrate the LI-COR response (since we were working at the low end of the mass flow controller (MFC) range) as well as the injection volumes. Most of the components used introduce negligible uncertainty including: LI-COR response (<1%), electronic pressure and mass flow control (~1% in most cases, see discussion below), preparation of solutions (at most 2% based on glassware and scales), true concentration of $CO_2$ in the cylinder used for calibration (<1%), and injection volume of syringes in most cases (<5%). The uncertainty in injection volume does not factor in when we are calculating ratios of FID and $CO_2$ peaks for a given injection. In contrast, integration of chromatographic peaks has a precision of roughly 10% (in most cases, see next comment), so the ratio of two integrated peaks (i.e., FID/$CO_2$) has a combined uncertainty of ~14%. Precision of all data in this manuscript is consequently on the order of 15%. Uncertainty in flows is not negligible for calibration of the $CO_2$ instrument, and thus in assessment of complete combustion; that issue is discussed below.

For an individual analyte, error is therefore dominated by the inherent uncertainty in the approach itself, which has typical uncertainties of ~20%. For bulk measurement of O/C, the uncertainty in the fit is <5%, so is negligible compared to the precision estimate of 15%. Based on AMS measurements referenced earlier, error in application of the fit to a mixture suggests a drop in error of 2-3x compared to individual analytes, which would indicate ~10%, again suggesting the 15% uncertainty in precision dominates uncertainty. Overall, this investigation suggests that an estimate of 15-20% is reasonable. We note that the presence of nitrogen in organic aerosol is also expected to yield a 10-20% uncertainty (see response to Reviewer #1, Comment #6), so does not substantially increase uncertainty even if it is not corrected. These levels of uncertainty are all comparable to the AMS, the most common current method for bulk aerosol O/C measurements.

(Section 2.2, lines 200-204; $CO_2$ error)

"Resolution on the $CO_2$ channel is slightly degraded due to band broadening during this transit, but chromatographic peaks are nevertheless clear. *The resolution provided by these detectors is sufficient for integration of the chromatographic peaks with ~10% uncertainty in most cases(Isaacman-VanWertz et al., 2017) (up to 20% for partially co-eluting peaks like those shown in Figure 2b).* Concentrations of $CO_2$ in the flow are only on the order of 1 ppm per 10 ng of analyte, resulting in the relatively low observed signal-to-noise on this channel."

(Section 2.4, lines 226-231; Calculation of FID/$CO_2$)

*"In System 2 and 3, individually injected analytes were normalized to injections of dodecane immediately preceding or following analysis. The selection of n-alkane for normalization is not critical, as nearly all saturated hydrocarbons were observed to have an FID/$CO_2$ ratio within 1% of the n-alkane average. The FID/$CO_2$ ratio is theoretically independent of the mass of analyte introduced since both FID and $CO_2$ scale*

*with analyte quantity. However, to minimize uncertainty, n-alkanes for normalization were introduced at concentrations similar to the analyte of interest, which should account for any potential non-linearity in detector response."*

(Section 2.4, lines 235-240, FID combustion efficiency)

"To assess combustion efficiency, a known mass flow of $CO_2$ gas (2 *± 0.2 sccm* of 1% $CO_2$ in balance air) was introduced to the desorption cell in the same location as the injection of analytes to provide a signal-to-mass response factor for the $CO_2$ analyser. *As all flows and pressures are controlled electronically, this flow of known calibrant undergoes the same flow and pressure conditions as any desorbed analyte. Uncertainty in calibrant flow (10%, due to operating the mass flow controller at the low end of its full scale) and $CO_2$ dominates over other sources of uncertainty (e.g. analyte mass injected) in this calibration."*

(Section 3.1, lines 267-280; Complete combustion)

"The average conversion of all analytes is *94±9 %, within error of complete combustion. This standard deviation is actually lower than the estimated 15% uncertainty in the amount of $CO_2$ measured (combined 10% uncertainty in calibrant flow and 10% uncertainty in peak integration); uncertainty in amount of carbon injected is comparatively lower, estimated as 5% uncertainty in solution concentrations and injection volumes.* Less oxygenated analytes (squalene and diethyl phthalate, introduced as solutions in DCM) exhibited efficient conversion with highly reproducible results: 95% conversion and a relative standard deviation (RSD) between replicate injections of ~5%. More oxygenated components, which were introduced as aqueous solutions, were more variable. Hydroxyethyl methacrylate ("HEM") had a mean conversion of 100%, but with a somewhat more variable RSD of 13%. Propylene glycol had a mean yield of only 87% and an RSD of 7%. *These data are tabulated in Supplementary Information Section S6.* These differences may be explained in part by solvent effects. The DCM could be evolved entirely before heating the cell, yielding higher precision for squalene and diethyl phthalate trials. However, solvent blanks of water gave small signals on the $CO_2$ detector and corrections were made to the HEM and propylene glycol peaks. As concentrations of HEM and propylene glycol became more dilute, the background water signal became comparatively large and uncertainty grew."

(Section 3.2, lines 289-298; Correlation between measured variables)

"FID/CO2 tends to be slightly lower than expected, which is likely due in part to uncertainty in structure-activity based estimation of ECN, which has been previously shown even for hydrocarbons to be on the order of 10% with a tendency to overestimate (Faiola et al., 2012) . Close correlations between FID/$CO_2$ and both rECN and O/C indicate that rECN and O/C must also be correlated, which is shown to be true in Figure 4c. Uncertainty in the average trends of these relationships is very low, with uncertainty in the *fitted* slopes of less than 4% in all cases *(uncertainty in all fit coefficients provided in Table S7)*. For 14 of the analytes shown, FID/$CO_2$ was measured in more than one instrument configuration, with results from one configuration always within 7% of the average value for an analyte (Figure S2). FID/$CO_2$ is therefore largely independent of the mechanism by which an analyte was thermally transferred, and

uncertainty in the measured $FID/CO_2$ of an individual component is on the order of *15%, in agreement with the more formal analysis of errors discussed in Section 3.3. "*

(Section 3.2, lines 326-331; Extrapolation to complex mixtures)

*"However, extrapolation to more complex mixtures is supported by both the theoretical principles of the approach, and previous work on measurement of O/C by the Aerosol Mass Spectrometer (AMS). That instrument is similarly calibrated as simply the average relationship between analyte O/C and the measured parameter (for the AMS, O/C of molecular fragments) for a large number of individual analytes (Aiken et al., 2008; Canagaratna et al., 2015). Canagaratna et al. found that uncertainty is actually highest in applying the average relationship to one or two components, and decreases with mixture complexity as the average relationship better describes the complex mixture (Canagaratna et al., 2015)."*

The TAG cell used here is an impactor and did not contain a filter (though a filter-based cell is used on some TAG systems for collection of semi-volatile gases). Introducing a filter would enhance uptake of semi-volatile gases, which might be an interesting application but is beyond the scope of the current project. While an impactor may also have some gas-phase adsorption, this could be removed by an upstream charcoal denuder, as discussed in our response to Referee #1, Comment #3 above. The considerations raised by the reviewer are good examples of the technical hurdles that would need to be addressed in applying the fundamental relationships presented here for use in a field-deployable instrument for measurement of particle O/C. The details of the TAG cell have been clarified (Section 2.2, lines 195-198):

*"The metal desorption cell used in Systems 2b and 3 was a passivated steel cell with an attached heater cartridge for temperature control, identical to that previously described for aerosol sampling and desorption using an internal impactor jet (removed for this work) as part of the field-deployable Thermal desorption Aerosol Gas chromatograph (TAG) (Kreisberg et al., 2009)."*

Page 6. Lines 165-166, and Figure 2b. It seems clear that there was incomplete separation in the GC peaks, especially for the CO2 detector. How were these analyzed and how did that impact the results e.g. accuracy and precision?

The reviewer is correct that on System 1, some analytes co-eluted from the GC. We in fact specifically chose to show the section of chromatogram in Figure 2b because it clearly shows the capability of resolution on the $CO_2$ detector, as well as the differences in resolution between the FID and $CO_2$ detectors. For analysis of co-eluting peaks, we used the TERN software package, which has been previously investigated in detail (Isaacman-VanWertz, 2017, Journal of Chromatography A, 1529:81-92, doi.org/10.1016/j.chroma.2017.11.005). In brief, error in the integration of co-eluting peaks is generally below ~20% as long as some "saddle" is visible between peaks (as in Figure 2b). This is somewhat higher

than the 10% typical precision for integration, but only applies to co-eluting peaks on System 1 and does not impact Systems 2 or 3.

The manuscript has been clarified (Section 2.2, Lines 199-204):

"Sample data from System 1 is shown in Figure 2b. FID response and $CO_2$ response are closely coupled, with a delay in the $CO_2$ signal of 3-5 seconds due to transit time through the permeation dryer. Resolution on the $CO_2$ channel is slightly degraded due to band broadening during this transit, but chromatographic peaks are nevertheless clear. *The resolution provided by these detectors is sufficient for integration of the chromatographic peaks with ~10% uncertainty in most cases (Isaacman-VanWertz et al., 2017) (up to 20% for partially co-eluting peaks like those shown in Figure 2b).* Concentrations of $CO_2$ in the flow are only on the order of 1 ppm per 10 ng of analyte, resulting in the relatively low observed signal-to-noise on this channel."

Page 6, Line 183. Doesn't normalizing to the nearest n-alkane create problems and uncertainties? It seems like some oxygenates might be in a completely different retention time range than the number of carbons it has. The carbon count could easily be off by one or more carbons.

We have clarified in the manuscript that the n-alkane used for normalization was that with the nearest retention time, not the same carbon number (e.g., 1-octanol, with retention index = 1078, was normalized to undecane, not octane). This is to account for any variability in instrument response as a function of time. However, practically speaking, there was almost no difference in the FID/$CO_2$ of hydrocarbons, so the conclusions are insensitive to which n-alkane is used for normalization.

 We have reworded the manuscript (Section 2.4, lines 225-226) as shown below:

"*In System 1, which involved GC separation, the*  FID/$CO_2$ of an analyte was normalized to that of the n-alkane with the nearest retention time."

Page 7. Section 3.1 I have a hard time believing that FIDs are not 100% efficient in converting carbon compounds to CO2: isn't this known? There are errors in both quantities plotted in Figure 3. Those need to be shown for the data points and other possible sources of systematic error should be discussed. For example, the CO2 instruments are concentration sensitive, so flow rates and pressures need to be known accurately and/or controlled so that they don't change as the experiments are being conducted.

We agree that an FID is designed to be 100% efficient under their designed flow conditions, though some inefficiencies may be possible at non-ideal hydrogen:air ratios, and thank the reviewer for pushing for a more careful examination of error in Section 3. We have also added a new section to the Supplementary section, S6, along with Table S6, shown below, that lists the average percent combustion and standard deviation of each compound. The combined data give a mean % yield and standard deviation of 94 ± 9 %, which is uncertainty of complete combustion.

In particular, we note that response produced by an FID is thought to occur through the formation of CHO*, and its subsequent relaxation to produce the CHO$^+$ ion, which ejects an electron that is measured by the instrument. This ion subsequently forms CO through reaction with water (Holm, 1999, Journal of Chromatography A, 842:221-227, doi:10.1016/S0021-9673(98)00706-7). In other words, efficient FID operation requires only conversion to CO, not necessarily $CO_2$. Any CO not converted to $CO_2$ would not harm operation of the FID, but would introduce error to the approach in this manuscript. Consequently,

we believe it is important to test the assumption of complete conversion to $CO_2$, and so find it to be approximately met within error.

The manuscript is revised as follows (Section 3.1, lines 263-281)

"Quantification of $CO_2$ produced from the analysis of known amounts of analytes provides an estimate of the efficiency of the conversion from organic carbon to $CO_2$ in the FID. *Though FIDs are designed for complete combustion, incomplete conversion to $CO_2$ due to e.g., incorrect hydrogen-to-air ratios, could result in high error or variability in measured FID/$CO_2$ ratios, particularly if combustion efficiency is related to molecular structure. Combustion completeness is investigated in* Figure 3, depicting results for four different analytes of varying degrees of oxygenation, along with a 1:1 line for reference. The average conversion of all analytes is *94±9 %, within error of complete combustion. This standard deviation is actually lower than the estimated 15% uncertainty in the amount of $CO_2$ measured (combined 10% uncertainty in calibrant flow and 10% uncertainty in peak integration); uncertainty in amount of carbon injected is comparatively lower, estimated as 5% uncertainty in solution concentrations and injection volumes.* Less oxygenated analytes (squalene and diethyl phthalate, introduced as solutions in DCM) exhibited efficient conversion with highly reproducible results: 95% conversion and a relative standard deviation (RSD) between replicate injections of ~5%. More oxygenated components, which were introduced as aqueous solutions, were more variable. Hydroxyethyl methacrylate ("HEM") had a mean conversion of 100%, but with a somewhat more variable RSD of 13%. Propylene glycol had a mean yield of only 87% and an RSD of 7%. *These data are tabulated in Supplementary Information Section S6.* These differences may be explained in part by solvent effects. The DCM could be evolved entirely before heating the cell, yielding higher precision for squalene and diethyl phthalate trials. However, solvent blanks of water gave small signals on the $CO_2$ detector and corrections were made to the HEM and propylene glycol peaks. As concentrations of HEM and propylene glycol became more dilute, the background water signal became comparatively large and uncertainty grew. Overall, however, the four compounds showed strong linearity and high percentage yields, supporting the conclusion that the *FID converts all analysed carbon to $CO_2$ without strong biases due to molecular structure.*"

Section S6 has been added to the Supplementary Information as given below:

**S6  Uncertainty in Complete Combustion Experiments**

*The data for the complete combustion experiments are presented in Table S6. As described in Section 3.1, squalene and diethyl phthalate were dissolved in dichloromethane, whereas hydroxyethyl methacrylate (HEM) and propylene glycol were dissolved in water. See also Figure 3 of the text.*

*Table S6.  Results for the complete combustion experiments*

| Compound | Number of points | Mean % yield ± Std Dev |
|---|---|---|
| Squalene | 23 | 95.2 ± 5.5 |
| Diethyl Phthalate | 23 | 95.7 ± 5.3 |
| HEM | 21 | 100.2 ± 12.8 |
| Propylene Glycol | 33 | 87.2 ± 7.4 |
| *All points* | | 93.7 ± 9.5 |

We have added to the manuscript a discussion of the errors on each point Figure 3 (see Section 3.1), but have chosen not to include them graphically, as the y-axis error bars would substantially overlap with each other, generally clutter the figure, and not substantially impact the interpretation of the figure. Error in the x-axis (ng of carbon injected) is dominated by the uncertainty in injection volume (estimated at 5%, filling a 2.0 μL microsyringe with 1.0 μL solution) with a smaller error anticipated in preparing the solutions, ~2%.  This would give a 5.4% uncertainty in the x-axis. For the y-axis, sources of uncertainty in the calibration include peak integration (~10%), error in the 1% $CO_2$ tank concentration (<1%), and error in the mass flow rate of $CO_2$. While most flows are controlled to better than a few percent, $CO_2$ calibration flow required the use of flows at the lower end of the mass flow controller range (i.e., 2 sccm controlled by a 100 sccm mass flow controller), yielded uncertainty of 10%. The combined uncertainty of peak integration and calibration flow results in a 14% uncertainty in the y-axis; this amount is approximately equal to the vertical spread of data points. These uncertainties further confirm that our data represent approximately complete combustion.

The caption for Figure 3 has been edited as follows:

"Figure 3: FID combustion efficiency, shown as ng of carbon measured by the $CO_2$ detector versus ng C injected as one of four analytes: squalene ($C_{30}H_{50}$), diethyl phthalate ($C_{12}H_{14}O_4$), hydroxyethyl methacrylate ($C_6H_{10}O_3$) and propylene glycol ($C_3H_8O_2$). *Uncertainty in the y-axis is approximately 15% and in the x-axis is <10%; error bars not included for clarity. Percent conversion for each compound and overall is tabulated in Supplementary Section S6.*"

As for systematic uncertainty, the main source of error is in peak integrations. All flow rates and pressures are controlled by electronic controllers so that variability is less than a few percent throughout operation

Page 7 Line 224. I had to read this several times to decide I basically disagree with this statement. Figure 1 suggests there will be considerable uncertainty in the relationship between rECN and O/C but is doesn't imply anything about FID/CO2. The extension to FID/CO2 is really because both carbonyl and carboxyl carbons have no FID response.

We are not totally clear on the reviewer's concern, but think it may arise from some confusing language in the original manuscript. The correlation between rECN and O/C is actually somewhat tangential to the reason that FID/CO2 is expected to correlate with rECN. We have tried to clarify the manuscript as below (Section 3.2,lines 284-289):

*"Figure 4 shows correlations between three parameters: FID/$CO_2$ signals, estimated relative ECN, and O/C. FID/$CO_2$ is the measured amount of FID signal generated per $CO_2$ produced, which is, assuming complete conversion of all carbon in an analyte, the amount of FID signal per carbon atom in the analyte. By normalizing this value to an n-alkane, FID/$CO_2$ provides a measure of the amount of FID signal generated per carbon atom in the analyte relative to a hydrocarbon, which is the definition of rECN. This observation suggests that rECN should equal the measured FID/$CO_2$, which are indeed observed to correlate closely (Figure 4b)."*

Page 8., Line 229. I this ±4% from the fit to the data?

Correct. This is the uncertainty in the fitted slope. Please see the uncertainties in the fits for Figure 4 given below on p.35-36.

Page 8., Line 258- Page 9, Line 259. This statement doesn't seem correct, especially when one considers compounds like oxalates, glyoxal, glycolic acid and the like. There is an inherent ±25% built into the technique.

(See Referee #1, Comment #4 above)

As discussed above, uncertainty is highest for individual analytes; as the mixture gets more complex, the average relationship tends to better describe the system. As noted about, there is indeed significant uncertainty in applying to the relationships to glyoxal and other carbonyl-heavy compounds (though oxalic and glycolic acids actually fall close to the line), but they are not expected to significantly bias bulk particle measurements. We note that our estimated uncertainty of ~20% for higher O:C is calculated roughly in the same way as that reported by the AMS (i.e., the average uncertainty of individual analytes).

Corrected.

Corrected.

Sorry for the confusion; our reference editing software somehow inserted a space, making it look like a typo. Fock's title and text do refer to "GLC".

Corrected.

Figure 4 – Please give the uncertainties in the fit parameters, i.e. slope, intercept

The fit parameters of the Figure 4 plots have been included in Supplementary section S7 and Table S7, as shown below:

*S7 Uncertainty in the correlations between rECN, O/C and FID/CO$_2$*

*Uncertainties for the slope and intercept of the three plots of Figure 4 are given in Table S7.*

*Table S7. Correlation parameters and uncertainties for Figure 4 plots*

| Figure | Plot | Slope ± error | % error | Intercept ± error | % error |
|--------|------|---------------|---------|-------------------|---------|
| 4a | FID/CO$_2$ vs O/C | -0.54 ± 0.02 | 3.7 | 0.98 ± 0.01 | 0.6 |
| 4b | FID/CO$_2$ vs rECN | 0.85 ± 0.03 | 3.8 | 0.14 ± 0.03 | 21.5 |
| 4c | rECN vs O/C | -0.60 ± 0.02 | 3.1 | 0.99 ± 0.01 | 0.6 |

The results are given below (but not in Section S7) with more significant figures.

For Figure 4a (FID/CO$_2$ vs O/C):  Slope = -0.53518 ± 0.0197 (3.7% uncertainty)

Intercept= 0.98842 ± 0.00583 (0.59% uncertainty)

For Figure 4b (FID/CO$_2$ vs rECN) :   Slope = 0.84675 ± 0.0324 (3.8% uncertainty)

Intercept = 0.14113 ± 0.0285  (21.5% uncertainty)

For Figure 4c (rECN vs O/C) : Slope = -0.60123 ± 0.0184  (3.1% uncertainty)

Intercept = 0.99458 ± 0.00543 (0.55% uncertainty)

Besides the added Supplementary section, S7, the caption of Figure 4 will report the % uncertainty in the slope and intercept as shown below:

Figure 4: Plots relating the three variables: measured FID/CO$_2$ relative to *n*-alkanes, relative ECN, and O/C. Comparisons shown are (a)  FID/CO$_2$ versus O/C, (b)  FID/CO$_2$ versus relative ECN, and (c) relative ECN versus O/C. Dashed lines are linear fits; fits assume error only in dependent variable in the case of comparisons to O/C (which has no error), and assume error in both variables ("orthogonal fit") in the case of rECN comparison to FID/CO$_2$. Methanol is shown in each plot as an unfilled marker as there are physical reasons it may be an outlier (discussed in the main text). *The respective percentage error of the slope and intercept, respectively, for each relationship are: (a) 4%, 1 % ; (b) 4 %, 22%; (c) 3%, 1%.*

Conclusions; The authors have failed to do the necessary work to demonstrate that this technique works for analyzing POC.

As discussed above, the focus of this manuscript is the validation of the O/C relationships with FID and CO$_2$ detector responses. Not all atmospherically-relevant applications of these relationships face the same uncertainties and issues, so tackling such issues for any one application warrants a dedicated manuscript demonstrating successful application. We believe our added section 3.4 ("Extension to atmospheric particles") makes this more explicit, as does our revised Introduction.

The whole paper needs to be re-cast to focus on gas phase species, or a considerable amount of additional work needs to be done on actual particle samples and the species that we already know they contain: N- and S- containing organic compounds.

We have added more discussion nitrogen and sulfur in both the Introduction and in Section 3.4. However, as noted throughout our response, we do not believe that the focus of this manuscript needs to be (or indeed should be) expanded to include the issues associated with tackling that particular application of this fundamental approach. We agree that using FID/$CO_2$ to estimate O/C for atmospheric particles does require building a system for sampling, possibly addressing sulfur and nitrogen (though as noted these are not expected to introduce major error), and likely comparing to existing instrumentation; all of which would have to be demonstrated to validate such an instrument. However, the basis of any such instrument would be relying on the relationships described in this manuscript and would need to validate the principle of operation as we have done in this manuscript. The focus and scope of this manuscript is to provide that underlying validation, so that future work can focus on demonstrating possible specific applications and overcoming specific technical hurdles of a given application. Extending to nitrogen- or sulfur-containing compounds may (and likely will) require inclusion of additional detectors (e.g., flame photometric detector for sulfur) that are unrelated to work in this manuscript. In this revision, we have worked to clarify the scope (and limitations) of this manuscript.

As far as the built-in ±25% uncertainty in O/C, the authors need to make a case that their measurement can still be useful in spite of this feature.

(Please see responses above to Referee #1, Comment #4 and Referee #2, Comment #11)

We thank the reviewer for pushing us to provide a more careful discussion of the uncertainty in our approach and have included a deeper discussion in the revised manuscript. As discussed in our responses to your general comment (page #26-27) above, the uncertainty of this approach is similar to (or in some cases slightly lower than) that of the AMS. Furthermore, we believe that the fundamental relationships described here have multiple possible applications (three are described in the Conclusions), including both bulk analysis of organic aerosol (for which our approach would likely have similar error as the AMS [Canagaratna et al., 2015, Atmos. Chem. Phys. 15:253-272] but with cheaper deployment) and analysis of individual analytes (for which our approach would allow substantially lower uncertainty in the calibration of some currently deployed instruments that suffer orders of magnitude uncertainty for some analytes [Isaacman-VanWertz et al., 2018, Nature Chemistry, 10:462-468]).

---

## Author Response (AR2)

**Response to Editor**

We are pleased that our manuscript has been accepted for publication following the corrections you suggested.

We agree with the Editor and Reviewer #3 that O/C is of general interest to atmospheric chemistry beyond particulate matter. Therefore we have revised the Abstract and Introduction to reflect a broadened discussion of reactive organic carbon and its properties.

The revised text is excerpted below, with the added text in *italics* and the line numbers provided.

Abstract (lines 10-29)

[revised manuscript text omitted]